


# Area-averaged evapotranspiration over a heterogeneous land surface: Aggregation of multi-point EC flux measurements with high-resolution land-cover map and footprint analysis

Feinan Xu[1,2], Weizhen Wang[1], Jiemin Wang[1], Ziwei Xu[3], Yuan Qi[1], Yueru Wu[1]

[1] Key Laboratory of Remote Sensing of Gansu Province, Heihe Remote Sensing Experimental Research Station, Northwest Institute of Eco-Environment and Resources, Chinese Academy of Sciences, Lanzhou, 730000, China
[2] University of Chinese Academy of Sciences, Beijing, 100049, China
[3] State Key Laboratory of Remote Sensing Science, School of Geography, Beijing Normal University, Beijing, 100875, China

*Correspondence to*: Weizhen Wang (weizhen@lzb.ac.cn)

**Abstract.** The determination of area-averaged evapotranspiration (ET) at the satellite pixel scale/model grid scale over a heterogeneous land surface plays a significant role in developing and improving the parameterization schemes of the remote sensing based ET estimation models or general hydro-meteorological models. The Heihe Watershed Allied Telemetry Experimental Research (HiWATER) flux matrix provided a unique opportunity to build an aggregation scheme for area-averaged fluxes. Based on HiWATER flux matrix datasets and a high-resolution land cover map derived from aircraft remote sensing, this study focused on estimating the area-averaged ET over a heterogeneous landscape with footprint analysis and multivariate regression. Firstly, the representativeness of multi-point eddy covariance (EC) flux measurements was quantitatively evaluated. The results show, the model estimated flux values cannot be directly validated with the flux tower measurements because the latent- and sensible heat fluxes measured by EC are determined by the upwind surface flux emanating from separate land cover classes, and a method in retrieving area-averaged fluxes should be applied. Secondly, a flux aggregation method was established combining footprint analysis and multiple regression analysis. The area-averaged sensible heat fluxes were obtained using the method and validated by the large aperture scintillometer (LAS) measurements.





Finally, the area-averaged ET of the kernel experimental area of HiWATER was estimated through the flux aggregation schemes. The aggregated results were then regarded as ground truth for the remotely-sensed ET products. These findings demonstrate that the refined flux integration technique is a better method to determine the heterogeneous surface fluxes.

## 1 Introduction

Land surface evapotranspiration (ET) is not only a key component in the regional water circulation, but also essential in the surface energy balances and land surface process. Under the condition of increasing shortage of water resources, high precision estimation of ET at regional scale is essential for those research fields, such as the management of basin water resources, regional planning and the sustainable development of agriculture (Wang et al., 2003). Currently, the commonly used methods for acquisition of regional ET are ground-based observation, remote sensing based estimation and model simulation, respectively.

The earth's surface is always characterized by spatial heterogeneity, and large land surface heterogeneity affects the exchange of momentum, heat, and water between the land surface and atmosphere (Mengelkamp et al., 2006). Indeed, the surface heterogeneity caused either by the contrast in soil moisture or vegetation type generates a large spatial variability of fluxes which limit the use of the eddy covariance (EC) system, unless one deploys a network of EC devices (Ezzahar et al., 2009b). Flux tower group can quantify the turbulent exchange of energy and mass between the atmosphere and a variety of surface types (Sellers et al., 1995), and these local point measurements need to be aggregated to provide a meaningful area averaged fluxes (André et al., 1986). If special aggregation rules for local flux measurements are applied, measurements can provide averaged fluxes at model grid scale (Beyrich et al., 2006;Mahrt et al., 2001). But given the EC network's high price and the requirement for their continuous maintenance, the large aperture scintillometer (LAS) is a useful alternative method for directly measurements of area-averaged sensible heat fluxes ($1-5$ km) (Ezzahar et al., 2009b;Ezzahar



and Chehbouni, 2009).

Satellite remote sensing have been considered as a promising data source for deriving regional ET data with the development of remote sensing technique (Ezzahar et al., 2009a). In response to increasing demand for spatially distributed hydrologic information, many satellite-based approaches have been developed for routine monitoring of ET at regional scale (Anderson et al., 2012). Nevertheless, the effectiveness of the satellite-based methods for estimating ET must be fully assessed by ground-based area-averaged flux measurements, due to the uncertainties of model inputs and parameterization schemes (Wang et al., 2003). Furthermore, there is a bias in directly comparing a satellite-based ET estimation with in-situ measurements, because of their spatial-scale mismatch and spatial heterogeneity at the sub-pixel scale (Jia et al., 2012).

General atmospheric-hydrological models (e.g., Numerical Weather Prediction) can adequately describe the interaction between the atmosphere and the underlying surface using complex parameterization schemes. The development and validation of these models are usually based on measurements performed over homogeneous land surfaces. While the assumption of homogeneity might be justified at the local scale (10 m – $10^3$ m), it is often violated at the scale of the grid resolution of current regional atmospheric models (about $10^4$ m) (Beyrich et al., 2006;Beyrich and Mengelkamp, 2006). Therefore, it is significantly important to determine the area-averaged surface fluxes at the satellite pixel scale/model grid scale ($10^3$ m – $10^4$ m) for the evaluation of general hydro-meteorological models and remotely sensed products.

A number of international field experiments have been performed over heterogeneous land surfaces in different geographical and climate regions of the earth in recent decades (Mengelkamp et al., 2006;Beyrich et al., 2006;Wang, 1999), such as HAPEX–MOBILHY (André et al., 1986), FIFE (Sellers et al., 1988), HAPEX-SAHEL (Goutorbe et al., 1994), BOREAS (Sellers et al., 1995), NOPEX (Halldin et al., 1998), LITFASS-2003 (Mengelkamp et al., 2006). In these experiments, based on multi-point flux measurements, surface fluxes at the model grid scale were obtained using various flux aggregation





techniques. The aggregated fluxes were also compared with those obtained from LAS systems and remote sensing estimation methods. The simple flux aggregation methods most commonly used in previous studies mainly include: arithmetic average method, the area-weighted method and the footprint-weighted method (Liu et al., 2016). These studies revealed, the combination of area-averaged flux measurements and multi-site flux measurements with simple flux aggregation schemes can provide reasonable estimates over a heterogeneous land surface (Mahrt et al., 2001;Beyrich et al., 2006;Liu et al., 2016).

However, the integration schemes of aforementioned methods assumed, the local flux measurements are representative of the individual surface type. This assumption can certainly lead to some errors, because the surface heterogeneity is also encountered at the field scale (Ezzahar et al., 2009c). To overcome this problem, footprint analysis can be an operational approach for the interpretation of tower flux measurements over a heterogeneous land surface (Schmid, 2002). The development of footprint models provides diagnostic tools to quantify the representative of tower flux measurements for selected sites (Horst and Weil, 1992;Kim et al., 2006). Besides, it had been demonstrated that the footprint climatology can be combined with information provided by satellite image (Kim et al., 2006;Chen et al., 2008). Land cover reflects the combined effects of vegetation, climate, soil and topography, some relationship should be expected between land cover and measured surface fluxes (Ogunjemiyo et al., 2003). Ran et al. (2016) proposed four indicators with footprint analysis and land-cover type map to improve the representative of EC towers and correct the EC flux measurements. But this method didn't obtain the surface fluxes of individual land cover types but just corrected the EC observations with some prior coefficients. Some previous studies have successfully related the aircraft observed fluxes to surface cover types with the combination of footprint models and remotely sensed land classification map (Ogunjemiyo et al., 2003;Kirby et al., 2008;Hutjes et al., 2010). Among these work, a flux dis-aggregation method (Hutjes et al., 2010), developed from former study presented by Ogunjemiyo et al. (2003), would be a promising method for integrate multiple tower-based



flux measurements to satellite pixel or grid scale. The application of this method in dis-aggregating heterogeneous EC flux measurements to separate land over classes have been not yet fully investigated.

A multi-scale observation experiment on evapotranspiration over a heterogeneous land surface was conducted in the middle reaches of Heihe River Basin during the HiWATER (Heihe Watershed Allied Telemetry Experimental Research) program in 2012 (Li et al., 2013;Liu et al., 2016). A comprehensive flux matrix, consisted of 18 EC systems and four groups of LAS systems within a $5 \times 5$ km$^2$ area, was specifically designed to capture the multi-scale characteristics of ET over a heterogeneous landscape during the experiment. HiWATER flux matrix, with an abundant of multi-scale flux measurements, provided a unique opportunity to build an aggregation scheme for area-averaged fluxes over a heterogeneous land surface. The objective of this study was to integrate multi-point EC flux measurements to area-averaged fluxes over a heterogeneous land surface with high resolution land-cover data and footprint analysis. The main issues were as followed: (1) the representative of EC flux matrix was quantitatively evaluated; (2) a flux aggregation scheme was established to estimate area-averaged sensible heat fluxes, taking LAS measurements as reference; (3) the area-averaged evapotranspiration, determined by the developed aggregation method, was regarded as a ground truth for remotely sensed ET products.

## 2 Study sites and data

### 2.1 Site description

This study was based on ground-based observation datasets, collected from the multi-scale flux matrix of HiWATER program from May to September 2012. The kernel experimental area ($5 \times 5$ km$^2$) of the multi-scale observation experiment was located in the Yingke and Daman irrigation district within Zhangye oasis. The land-cover types were dominated by maize (72 %), vegetables (5 %), orchard and shelterbelt (8 %), residential area and roads (15 %). As shown by the numbers $1 - 17$ in the following Fig. 3, 17 sites were installed according to the distribution of crop planting structure and land





cover. Each of them was equipped with an eddy covariance system (with two layers in site 15) and an automatic weather station (AWS), to capture the exchange process of surface water and energy budget at the local scale and micrometeorological elements near the surface layer. Spatial distribution of EC/AWS systems is shown in Fig. 3, with site 1 of vegetable (pepper) field, site 4 of residential area, site 17 of apple orchard, and the others are in maize fields. Key micrometeorological measurements observed at each AWS included four-component radiation, one or two levels wind / temperature / relative humidity profile, soil temperature / moisture and soil heat flux, etc. Among these sites, site 15 was a superstation equipped with two levels EC system, and seven-level wind speed/direction, air temperature/humidity profiles. 4 paths of large aperture scintillometer devices were installed crossed over the experimental district to obtain area-averaged sensible heat fluxes (see Fig. 3). Details of the EC and LAS systems in the flux matrix were given in Table 1 and Table 2, respectively.

## 2.2 Data collection and processing

### 2.2.1 Flux data processing

Data in typical clear days of 29 to 30 June 2012 were selected for the following analysis, including EC data from 16 towers (except site 3 and highest level (34 m) of site 15) and LAS data as well as multi-point micrometeorological data list above. During the two days, there was almost no irrigation in the flux matrix. At first, AWS data sampled at 10 min were averaged to 30 min period. Secondly, careful data processing and quality control for EC and LAS raw data were performed to obtain high quality flux data.

The EddyPro software (www.licor.com/eddypro) developed by the American LI-COR company was used to process and calculate the 10 Hz raw EC data into a half hour averaged-flux data with flag (0, 1, 2) by several steps, including spike removal, time lag correction, coordinate rotation (2-D rotation), frequency response correction, sonic virtual temperature correction, and corrections for density fluctuation (Webb-Pearman-Leuning, WPL). And a quality assessment was performed using the flag





system according to both the stationarity tests and turbulent development tests. Detailed information on the processing steps of EC raw data can be found in Wang et al. (2015) and Xu et al. (2013).

The flux data of flag 2 were discarded, as well as the night data when the friction velocity was below 0.1 m s$^{-1}$ (Blanken, 1998;Liu et al., 2011). To obtain daily ET, a gap-filling method based on the nonlinear regression (establishing the relationship between latent heat flux and the net radiation) was used to fill the gaps between the 30 min latent heat flux data. Finally, the daily ET was calculated by summing the half-hourly gap-filled ET to 24 h totals.

The LAS system provided a measurement of the structure parameter for the refractive index of air ($C_n^2$) with an output period of 1 min, the raw LAS data were, firstly, averaged to 30 min (the LAS data only observed by the BLS series were collected). Then, the path-average sensible heat fluxes were iteratively calculated combining EC data (e.g. length of stability and Bowen ratio) and meteorological data (e.g. wind speed, air temperature, pressure) based on Moninin—Obukhov Similarity Theory (MOST). To perform the quality control for raw LAS data, the equation $C_n^2 < 0.193L^{8/3}\lambda^{1/3}D^{5/3}$ (L is the path length, D is the diameter of optical aperture, and $\lambda$ is wavelength) was applied to remove the data whose value exceeded the saturated criterion (Ochs and Wilson, 1993). Only sensible heat fluxes from LAS measurements for daytime conditions (8:30 am − 15:30 pm, Beijing Standard Time) were selected in this study.

### 2.2.2 Collection and processing of remote-sensing data products

The airborne hyper-spectral images over the kernel experimental area of the HiWATER were acquired by the Compact Airborne Spectrographic Imager (CASI) on 29 June 2012. The collected hyper-spectral images had 48 bands that ranged from 380 nm to 1050 nm in wavelength, with the spectral resolution and spatial resolution of 7 nm and 1 m, respectively. The atmospheric correction and geometric rectification of the CASI data were carefully conducted with the ground control points measured simultaneously (Xiao and Wen, 2013). The LiDAR data were acquired by an airborne LiDAR



system on 19 July 2012, the average point density of the data was 4 points per m$^2$ (Xiao and Wen, 2014). After several steps of processing, the Canopy Height Model (CHM) data were derived from the LiDAR data. Finally, the CASI hyper-spectral images and CHM data were combined to map the high resolution land cover type with an object-based classification method. The classification accuracy of the 1-m land cover map is higher than 90 %, and Kappa coefficient is approximately 0.9. More detailed information on the classification of the map can be found in Liu and Bo (2015).

However, there still occurred land cover misclassification in the collected map, despite high accuracy of the land cover product. One of the reasons is the spectral similarity between different surface cover types. To obtain a much more accurate land cover map, the misclassified patches of land cover in the kernel experimental area were visually and manually revised based on high-resolution CCD images and Google Earth imagery acquired in 2012. Finally, for the aim of this study, the refined 12 kinds of land classification types in the study area were merged into four kinds (maize, vegetables, woods and non-vegetation types) according to crop species and surface types. As shown in Fig. 3 in the Sect. 4.2. Among them, non-vegetation types contain two types of surface cover, namely buildings and road.

In addition, the daily evapotranspiration of a subset of the kernel experiment area (yellow line in Fig. 3) during the intensive observation period was estimated from 1 m airborne CASI data by performing a modified Penman—Monteith (P-M) formula, based on observed meteorological data and a certain assumptions. The estimated results were in agreement with the EC measured values. See Qiao et al. (2015) for details on modified P-M method formulations and inputs requirements.

## 3 Methodology

### 3.1 Aggregation method combining footprint analysis and multivariate regression

It is general accepted that an average flux equals the area-weighted sum of the component fluxes emanating from individual land cover classes (Hutjes et al., 2010).





$$F = \sum_{k=1}^{n} A_k F_k \tag{1}$$

Where $F$ is the total flux of any scalar (here the heat and water vapor flux are on the study) for a specified area, $A_k$ is the fractional coverage of an individual land cover class $k$ within that area, and $F_k$ is the flux emanating from that individual land cover class, $n$ is the number of land cover classes that is distinguished in the specified area.

Then, the observed flux ($F_{obs}$) at height $z_m$ can be closely related to the true surface flux upwind of measurement point through the footprint function, in continuous form:

$$F_{obs}(x_{obs}, y_{obs}, z_m) = \int_{-\infty}^{\infty} \int_{-\infty}^{\infty} F(x, y, 0) w(x, y, z_m) dxdy \tag{2}$$

In Eq. (2), $x_{obs}$, $y_{obs}$ are the site coordinates. $z_m$ is the effective observation height defined as $z_m = z - d$, $z$ is the sensor height, $d$ is the zero plane displacement. The footprint $w(x, y, z_m)$ describes the flux portion seen at $(x_{obs}, y_{obs}, z)$. The Equation (2) can be discretized for a uniform grid over a landscape, as in a satellite image based land-cover map, leaving out the height dependence for simplification. Equation (2) becomes:

$$F_{obs} = \sum_{k=1}^{n} F_k \sum_{i=1}^{N} \sum_{j=1}^{M} w_{ij} \Delta x \Delta y \tag{3}$$

Where each pixel $\Delta x \Delta y$ of the map is assumed to be homogeneous, which is uniquely classified as belonging to class $k$. Then the fraction X of the $k$-th land cover type in the footprint (*fp*) was defined as:

$$X_{fp,k} = \sum_{i=1}^{N} \sum_{j=1}^{M} w_{ij} \Delta x \Delta y \tag{4}$$

Combing the Eq. (3) and Eq. (4), the multi-linear model for the flux becomes:

$$F_{obs} = \sum_{k=1}^{n} F_k X_{fp,k} \tag{5}$$

A critical assumption under the flux aggregation method is that each land cover $k$ (area $A_k$) is with a





constant source strength ($F_k$). Then Flux $F$ for a specific area is a weighted aggregation of its various land cover classes. Base on multi-point tower flux measurements, multi-linear regression equations can be formulated by overlaying the flux footprint with high resolution land cover map (Eq. 5). The equations could be solvable to get $F_k$ with the Least Squares method, when the number of flux towers is greater than that of land cover classes (n). For each LAS path, the observed (sensible heat) flux can be dis-aggregated by relevant footprint function as Eq. (5). This can be taken as a validation of the former step.

The accuracy of this aggregation technique is highly dependent on four aspects: (1) better flux data for all EC sites; (2) better land cover classification map; (3) more precise flux footprint analysis; (4) good flux and footprint data for LAS. So properly processed flux measurements, accurate high-resolution land cover map and appropriate footprint functions are the fundamental of formulating a better multiple linear regression. Sometimes, the established multi-linear regression equations may not be solved because of the adoption of low accuracy land cover classification map or imperfectly known footprint model. When suffered this problem, the classification accuracy of the used land cover map should be checked, and the selected footprint model should be verified whether it's applicable.

## 3.2 Footprint models

The Eulerian analytical footprint model, which developed by Kormann and Meixner (2001), was used to estimate the single time flux footprint of EC measurements. This footprint function $w(x, y, z)$ is composed of the crosswind integrated flux footprint function $f^y(x, z)$ and the Gaussian crosswind distribution function $D_y(x, y)$. The footprint function equation is followed by Eq. (6). More details on the mentioned parameters can be seen in Kormann and Meixner (2001).

$$w(x, y, z) = f^y(x, z) \bullet D_y = \frac{1}{\Gamma(\mu)} \frac{\xi^\mu}{x^{1+\mu}} e^{-\xi/\mu} \bullet \frac{1}{\sqrt{2\pi}\sigma} e^{-\frac{y^2}{2\sigma^2}} \qquad (6)$$

The flux contribution source area of the LAS measurements can be assessed by combining the





footprint model for point measurement with the path-weighting function $W(x)$ of the LAS (Meijninger et al., 2002). For equal sized transmitter and receiver apertures, this path-weighting function is symmetrical bell-shaped having a center maximum and tapering to zero at the transmitter and receiver end. The equation of footprint function of LAS is that:

$$f_{LAS} = \int_{x_2}^{x_1} W(x) \bullet w(x - x', y - y', z_{LAS}) dx \qquad (7)$$

Where $x_1$, $x_2$ are the positions of the LAS receiver and transmitter, respectively. $x$, $y$ represent the locations of points along the optical path length of the LAS. $x'$, $y'$ are the coordinates upwind of each of the points. $z_{LAS}$ is the effective height of the LAS measurements.

To obtain the daily flux footprint of the EC flux measurements, the flux-weighted footprint climatology method was applied for each pixel (Liu et al., 2016). The expression of footprint climatology function is shown in Eq. (8).

$$w_c(x, y, z) = \sum_i^N w_i(x, y, z) \bullet Flux(i) \bigg/ \sum_i^N Flux(i) \qquad (8)$$

Here $i$ denotes the timestep (e.g. 30 min), $N$ is the total number of 30-min periods within the time frame (e.g. daily), $Flux(i)$ is the observed flux at $i$ time-step (e.g. ET for every 30-min in this study), $w_i(x, y, z)$ represents every half-hourly footprint calculated by Eq. (6).

The inputs of the footprint models mainly include the measurement height, wind direction, wind speed and the Obukhov length. The daily flux contribution area of the EC flux measurements was calculated by Eq. (8), which provides approximately 90 % of the total source area that contributes to the measured fluxes. Every 30-min flux source area of the LAS sites was estimated via Eq. (7), and the 90 % half-hourly footprint contours of LAS measurements were used. The normalized daily and half-hourly footprint estimates were overlaid with 1-m land cover map to determine the footprint-weighted land cover contribution for EC and LAS sites.

**3.3 Framework of the determination of area-averaged fluxes**

The overall framework for determining the area-averaged evapotranspiration over a heterogeneous land surface mainly includes three aspects (Fig. 1).

Firstly, the spatial representativeness of the 16 EC flux towers within the $5 \times 5$ km$^2$ experiment area was quantitatively assessed by overlaying in-site flux footprint climatology with 1-m land cover map. Detailed analyses on this aspect are going to be presented in the following section.

The second aspect was to evaluate the reliability of the established flux aggregation schemes. The land cover specific flux was firstly dis-aggregated from multiple EC flux measurements by performing a multiple linear regression analysis (Eq. 5). The EC dis-aggregated fluxes of each land cover classes were then aggregated again according to the fractional weight of each land cover class in the LAS footprint (Eq. 4). Finally, the aggregated fluxes were compared with LAS observations.

At last, the area-averaged evapotranspiration over a heterogeneous land surface was estimated from multi-point EC flux measurements with the developed flux integration schemes, based on footprint models and high resolution land cover map. The estimates were used to validate the remotely sensed ET products.

## 4 Results and Discussion

### 4.1 The characteristics of the surface heat and water vapor fluxes

Understanding the characteristics of the surface heat and water vapor fluxes over different land surface types is fundamental to integrate the multi-point local scale flux measurements to area-averaged fluxes at satellite pixel or model grid scale through the flux aggregation schemes.

Figure 2a depicts the diurnal cycle of the sensible heat fluxes at different sites for two clear days, and the daily ET are shown in Fig. 2b. Both of the two figures reveal significant differences in the magnitude of the sensible- and latent heat fluxes between different surface types during the growing season.

The sensible heat fluxes over residential area reached a maximum of about 150 W m$^{-2}$ at afternoon

(H_ec4, Fig. 2a). On the contrary, it showed minimum daily ET values for Site 4, with approximately 3 mm − 4 mm during the two days, due to a certain fraction of sealed land surfaces (Fig. 2b).

Over the vegetated surfaces (orchard, vegetable, maize), nearly all the sensible heat fluxes were less than 100 W m$^{-2}$ (Fig. 2a). The sensible heat fluxes over different types of vegetation were also significant different, however, the magnitude of the differences between maize fields was relatively small.

Deviations were also found between daily ET over different vegetation types (Fig. 2b). This can be partly explained by the discrepancy in plant physiology and vegetation growing stage. The maize fields performed highly daily ET values and lower sensible heat fluxes, and significant variations in daily ET were found among the maize sites (Fig. 2b). It could be noticed, the divergences were closely related to the variability of sensible heat fluxes.

The preliminary results indicated that, the variance of the surface energy fluxes between the HiWATER tower flux sites was significant during the crop growth period. The differences in surface heat and water vapor fluxes between maize sites also could be noted.

## 4.2 Analysis of the representativeness of the multi-point EC flux measurements

To further understand the variability of surface energy fluxes between different sites in a heterogeneous landscape, the footprint analyses for representativeness of EC sites were applied by overlaying flux footprint with high resolution land-cover map (Fig. 3). The fraction of land cover classes present in the daily-averaged footprint of each EC measurements is given in Fig. 4. Given EC footprints boundary exceeded the extent of land cover map, sites 5, 8, 13 and 16 were not used for footprint analysis and not shown in Fig. 4.

Due to the variations in the observation height, atmospheric stability, wind direction and wind speed, the exact shape and size of the EC source area were distinctly different (Fig. 3). For each EC flux measurements, there was more than one type of land cover in its footprint. The contribution of each land





cover classes to the total measured flux for EC sites was changed with the varying source area (Fig. 4).

The dominated surface types in the source area were vegetable and orchard at sites 1 and 17, respectively. For site 4, however, there were mainly three types of land cover within its footprint, namely non-vegetation, maize and woods type. The fractional weight of the non-vegetation type and maize field in the footprint greatly varied, while the proportion of woods was almost changeless.

At maize field sites, the relative contribution of maize field to the EC measured flux was approximately more than 0.9, except for sites 2, 9 and 10. At site 2, the percentage of non-vegetation type in the footprint was almost 0.18. For site 9, the rate of maize and non-vegetation type present in footprint significantly varied. The contribution of vegetable type to the flux measurements at site 10 ranged from 0.15 to 0.1.

The above analysis shows, the latent- and sensible heat fluxes measured by EC systems are representative of the averaged fluxes, which were weighted the upwind surface flux emanating from individual land cover classes with flux footprint. In general, it may be problematic to validate the model estimated fluxes by direct comparison with tower-based flux measurements over a heterogeneous land surface. Thus, the extension of multiple flux measurements to pixel/grid scale surface fluxes is urgent.

## 4.3 Evaluation of the EC aggregated fluxes

The determination of area-averaged fluxes from point measurements is usually not straightforward, especially for heterogeneous land surfaces. Based on multi-point EC flux measurements and accurate 1-m land cover map, a flux aggregation method was established to estimate averaged surface fluxes with footprint analysis and multivariate regression. Fig. 3 shows that all types of land covers present in the LAS flux footprint, so the LAS measurements can be taken as references to assess the feasibility of the developed integration schemes.

At first, the sensible heat flux for each land cover was dis-aggregated from the EC observed component fluxes in a heterogeneous footprint with multiple linear regression method. The diurnal





cycle of the EC dis-aggregated sensible heat fluxes for each land cover types is highly significant (Fig. 5). During the crop growth stage, the daytime sensible heat fluxes for non-vegetation type exhibited a maximum of about 200 W m$^{-2}$. On the contrary, the maize field showed a minimum value in the afternoon. The sensible heat fluxes for forest and woods types lied between them.

5      Then, the EC aggregated sensible heat flux representative for the LAS source area was calculated by weighting the specific fluxes for the four land-cover classes with their relative fraction in the LAS source area. Fig. 6 illustrates a scatterplot of 30-min averaged sensible heat fluxes estimated by the flux aggregation method (hereafter referred as H_ECagg) versus LAS measurements (H_LAS), as well as the linear regression parameters. The different statistics between LAS observed fluxes and EC

10   aggregated results are listed in Table 3.

     For LAS 1 (see Fig. 6a and Table 3), a good agreement is found between EC aggregated fluxes and LAS measurements, with high correlation coefficient and low RMSE value ($R^2$= 0.79, RMSE= 0.96 W m$^{-2}$). The scatter points in the graph are nearly close to the 1:1 line. The MBE and MAPE values between aggregated fluxes and LAS observations were 4.25 W m$^{-2}$ and 9.93 %, respectively.

15      Compared with LAS 1, there was a little scatter between LAS measured fluxes and estimates from multiple EC flux observations for LAS 2, but yielding a small mean bias error (MBE = 2.31 W m$^{-2}$) (Fig. 6b, Table 3). RMSE and MAPE values between H_ECagg and H_LAS2 were much higher than that of LAS 1, with values of 6.91 W m$^{-2}$ and 16.39 %, respectively. Considering the heterogeneous distribution of surface covers, slight area of non-vegetation distributing in the center of LAS 2 path

20   would be the primary factor attributing to the bias (Fig. 3).

     For LAS 3 (Fig. 6c, Table 3), there was a slightly weak relationship between sensible heat fluxes derived from the LAS measurements and flux aggregation method, with correlation coefficient ($R^2$) of 0.57 and RMSE, MAPE and MBE values of 17.63 W m$^{-2}$, 31.7 % and -18.01 W m$^{-2}$, respectively. The scatter points in Fig. 6c were overall below the 1:1 line. It indicated that the 30-min averaged H

25   estimated from multiple EC flux observations were underestimated against LAS derived H (negative





MBE). As shown in Fig. 3, there is more large area of residential areas randomly distributing in the center of LAS 3 path than other three LAS systems. This discrepancy is likely related to the heterogeneously distributed surface types.

In Fig. 6d, the area-averaged sensible heat fluxes obtained using the flux aggregation method were consistent with LAS measurements, with $R^2$ of 0.57 for LAS 4. In contrast with LAS 3, the scatter points in this graph were almost above the 1:1 line (overestimate of EC estimated H, MBE > 10 W m$^{-2}$). RMSE value of LAS 4 relatively decreased by 4.88 W m$^{-2}$, but MAPE value was up to 33.7 %. The red open squares in Fig. 6d are more close to the 1:1 line than the blue open circles. Moreover, the main wind direction on 29 June was southeast wind, while northwest wind prevailed on 30 June 2012. The findings indicate that the magnitude of divergences between the estimated and measured area-averaged surface fluxes is in large part concerned with the variation of corresponding LAS source area (Fig. 1).

The above results demonstrate that the area-averaged fluxes, aggregated from multiple EC flux measurements with footprint analysis and high resolution land cover map, are reliable compared with the averaged fluxes measured by LAS. Therefore, the developed flux integration schemes in this study can be an effective way to estimate areal averaged fluxes.

## 4.4 Estimation of area-averaged evapotranspiration

To determine the area-averaged ET from multi-point EC flux measurements, the flux aggregation method combing footprint analysis and multivariate regression was performed with high resolution land-cover map.

Same as Sect. 4.3, the daily ET for each land cover classes was firstly separated from the multiple EC flux measurements with 1-m land cover map and daily-averaged flux footprint. The EC dis-aggregated daily ET for all the land covers over two clear days was shown in Fig. 7. As can be seen, the daily ET values for maize field were highest during the crop growing season (7 mm – 8 mm). The values of daily ET were 6.4 mm for woods type, and it ranged from 6 mm to 7 mm for vegetable type.





On the contrary, the daily ET for non-vegetation type varied largely, with 2.8 mm on 29 June, and 1.5 mm on 30 June.

The EC-disaggregated daily ET for all land cover classes was aggregated with 1-m resolution land cover map to map the spatial distribution of daily ET in our case study area. Fig. 8 depicts the spatial pattern of daily ET on 29 and 30 June 2012. It can be seen from the legend in figure, the daily ET ranged from 1.56 to 7.95 mm during the two days, and with higher values on 29 June (Fig. 8a) for all land cover classes than that on 30 June (Fig. 8b). The maize field performed highest ET value and distributed widely, whereas other three types of land cover randomly distributed across the whole study area with quite different ET values.

Table 4 lists the total ET for the different land cover classes and their proportion of the total area ET. The results demonstrated that the ratio of ET for maize field to the total area ET was in excess of 80 %. This finding further illustrate that the total ET of our study area was dominated by maize field. In addition, the total rate of ET for both woods and vegetables types was approximately 13 %, and the ET value for non-vegetation type accounted for 4.83 % of daily totals on the average.

To test the performance of model parameterization schemes and inputs, the EC aggregated ET maps were regarded as a validation of the remotely sensed products with 1-m spatial resolution.

A comparison of remotely sensed ET data with area-averaged ET estimated from a flux aggregation method using multiple EC flux measurements is listed in Table 5. Compared with EC aggregated results, the mean daily ET derived by modified P-M method was underestimated on the two days, with values of 1.64 mm and 1.12 mm, respectively. The daily ET values for vegetated surfaces (maize, woods and vegetables) which were estimated from airborne remote sensing data were lower than that obtained from EC aggregated method (about 1 mm). However, remote-sensing based estimation of ET for non-vegetation types was overestimated about 1 mm on 30 June.

The total ET values for each land cover classes derived from two estimation methods were summarized in Table 6. The statistical results showed, on the two clear days, the ratio of the



underestimate amounts of the remotely sensed ET values to EC aggregated results were 24.58 % and 17.71 %, respectively (the last row in Table 6). By comparison, remote sensing estimation of ET from vegetated land surface was largely underestimated, and the proportion of underestimation amount was more than 20 %. However, for non-vegetation type, the total ET on 30 June estimated from modified P-M model using airborne data was greatly overestimated, with the ratio of overestimate amount up to 63 %. This caused by the overestimation of ET for field roads, which value was close to the maize field.

All in all, the results demonstrated that the ET derived from airborne remote sensing data was greatly underestimated, compared with EC aggregated results obtained from multiple EC flux measurement. This finding indicates that the inputs and parameterization schemes in the modified Penman—Monteith (P-M) formula should be further optimized and developed.

## 5 Summary and conclusion

On the basis of 1-m accurate land cover map and multi-point ground-based flux measurements datasets from 16 EC systems and 4 groups of LAS systems during the intensive observation period of HiWATER program, the area-averaged surface fluxes over a heterogeneous surface were determined by a flux aggregation method established through the integration of footprint analysis and multiple regression, and validated by the LAS measurements to assess the reliability of the method. Ultimately, the integration method was used to estimate area-averaged ET for the verification of the remotely sensed ET products.

First and foremost, analyses of the spatial representativeness of multiple EC flux towers were performed for the interpretation of the surface fluxes over different land surfaces. It is proved in this paper that the combination of footprint analysis and high-resolution land cover map can be a proper way to clarify the relationship between the tower-based flux observations and individual land cover specific fluxes, and it is also the foundation for the establishment of flux aggregation scheme.

Secondly, based on good multi-scale (EC & LAS) flux datasets, precise flux footprints of flux

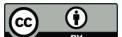



towers and better land cover classification map, a flux aggregation scheme can be successfully established through the integration of footprint analysis and multivariate regression. For a heterogeneous study area, the surface flux emanating from individual land cover classes need to be acquired by the developed flux integration method. The results show that the method provides a unique opportunity to disentangle the heterogeneous land surface fluxes in their single components.

Then, the averaged surface fluxes estimated from the developed flux aggregation method were compared with the corresponding observed LAS values. The MAPE values of the LAS 1, LAS 2, LAS 3 and LAS 4 were 9.93 %, 16.39 %, 31.7 % and 33.7 %, respectively. The results indicate that the divergences between EC aggregated estimates and LAS observed fluxes are much more significant at heterogeneous sites. one of the reasons is that the application of the flux aggregation method in this study did not take into account the information with respect to the soil properties (temperature, moisture, etc.) and the status of vegetation growth (NDVI, etc.). On the other hand, the uncertainties of the LAS observations, partly attributed to theoretical and personal reasons (e.g. variable Bowen ratio), lead to the inconsistency of EC-estimated and LAS-measured averaged fluxes, as well as the complex LAS footprints.

In spite of the limitations mentioned above, the flux integration technique refined in this study is feasible for the estimate of area-averaged fluxes over a heterogeneous land surface. Therefore, the area-averaged ET derived from the aggregation method can be taken as a ground truth for the remotely sensed ET products to verify the performance of the parameterization schemes in the remote sensing based models.

The results of this study also suggest, the dis-aggregation process that attribute EC observed fluxes over heterogeneous land surface to separate land cover classes has the potential to scale up multiple EC measurements to a landscape, even to a whole river basin through further studies, especially evapotranspiration. The implication of this result is not only greatly important for improving the parameterization schemes of surface fluxes in meso-scale (1 ~20 km) models but quite interested for



hydrological modeling and basin water resource management.

## Data availability

The flux observation matrix datasets from the eddy covariance (EC) systems and large aperture
scintillometer (LAS) systems and the meteorological data in this study are available at
http://card.westgis.ac.cn/hiwater/mso on request. The revised 1-m land cover data for this paper are
available from the corresponding author on request.

## Competing interests

The authors declare that they have no conflict of interest.

*Acknowledgements.* This study was supported by the National Natural Science Foundation of China
(Grant number: 41271359), the Key Project of National Natural Science Foundation of China (Grant
number: 41301363).

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



**Table 1** Details of the Eddy covariance systems in the HiWATER flux matrix

| Site No. | Longitude (°) | Latitude (°) | Elevation (m) | Sensor height (m) | Surface type |
|---|---|---|---|---|---|
| 1 | 100.3582 | 38.8932 | 1552.75 | 3.8 | Vegetables |
| 2 | 100.35406 | 38.88695 | 1559.09 | 3.7 | Maize |
| 3 | 100.37634 | 38.89053 | 1543.05 | 3.8 | Maize |
| 4 | 100.35753 | 38.87752 | 1561.87 | 4.2/ 6.2 after 19 Aug. | Residential area |
| 5 | 100.35068 | 38.87574 | 1567.65 | 3.0 | Maize |
| 6 | 100.3597 | 38.87116 | 1562.97 | 4.6 | Maize |
| 7 | 100.36521 | 38.87676 | 1556.39 | 3.8 | Maize |
| 8 | 100.37649 | 38.87254 | 1550.06 | 3.2 | Maize |
| 9 | 100.38546 | 38.87239 | 1543.34 | 3.9 | Maize |
| 10 | 100.39572 | 38.87567 | 1534.73 | 4.8 | Maize |
| 11 | 100.34197 | 38.86991 | 1575.65 | 3.5 | Maize |
| 12 | 100.36631 | 38.86515 | 1559.25 | 3.5 | Maize |
| 13 | 100.37841 | 38.86076 | 1550.73 | 5.0 | Maize |
| 14 | 100.3531 | 38.85867 | 1570.23 | 4.6 | Maize |
| 15 | 100.37223 | 38.85555 | 1556.06 | 4.5/ 34 | Maize |
| 16 | 100.36411 | 38.84931 | 1564.31 | 4.9 | Maize |
| 17 | 100.36972 | 38.8451 | 1559.63 | 7.0 | Orchard |



**Table 2** Details of the Large Aperture Scintillometers in the HiWATER flux matrix

| Site | Longitude (°) | Latitude (°) | LAS type | Path length(m) | Effective height (m) |
|---|---|---|---|---|---|
| LAS 1 | North 100.35090 | 38.88413 | BLS900 | 3256 | 33.45 |
| | South 100.35285 | 38.85470 | RR9340 | 3256 | 33.45 |
| LAS 2 | North 100.36236 | 38.88256 | BLS900 | 2841 | 33.45 |
| | South 100.36171 | 38.85717 | BLS450 | 2841 | 33.45 |
| LAS 3 | North 100.37319 | 38.88338 | BLS900 | 3111 | 33.45 |
| | South 100.37223 | 38.85555 | Kipp&zonen | 3111 | 33.45 |
| LAS 4 | North 100.37841 | 38.86076 | BLS450 | 1854 | 22.45 |
| | South 100.36840 | 38.84682 | RR9340 | 1854 | 22.45 |




**Table 3** Different statistics between LAS observed flux and EC aggregated flux at LAS sites

| LAS sites | RMSE [W m$^{-2}$] | MBE [W m$^{-2}$] | MAPE [%] |
|-----------|-------------------|------------------|----------|
| LAS1 | 0.96 | 4.25 | 9.93 |
| LAS2 | 6.91 | 2.31 | 16.39 |
| LAS3 | 17.63 | -18.01 | 31.70 |
| LAS4 | 12.75 | 10.66 | 33.70 |

Remarks: $RMSE=\sqrt{\sum_{i=1}^{n}(P_i-O_i)^2/n}$ , $MAPE=\frac{100}{n}\sum_{i=1}^{n}\frac{|P_i-O_i|}{\overline{O}}$ , $MBE=\sum_{i=1}^{n}(P_i-O_i)/n$ , $P_i$ is EC aggregated value,$O_i$ is LAS observed value,$\overline{O}$ is the

mean measured value, $n$ is the number of samples. RMSE is root mean square error, MAPE is mean absolute percentage error, MBE is the

mean bias error.



**Table 4** ET for each land cover classes and their proportion of the kernel experimental area ET

| Land cover class | Area [km$^2$] | 2012/06/29 | | 2012/06/30 | |
|---|---|---|---|---|---|
| | | ET [m$^3$ d$^{-1}$] | ET proportion of total ET [%] | ET [m$^3$ d$^{-1}$] | ET proportion of total ET [%] |
| Maize | 17.42 | 138434.21 | 81.61 | 127241.13 | 83.20 |
| Woods | 1.96 | 12775.93 | 7.53 | 12587.64 | 8.23 |
| Vegetables | 1.20 | 8275.22 | 4.88 | 7484.52 | 4.89 |
| Non-vegetation | 3.62 | 10141.41 | 5.98 | 5635.12 | 3.68 |





**Table 5** Comparison of the daily ET estimated from remote sensing data and multiple EC flux measurements (unit: mm d$^{-1}$)

| Date | Method | Mean daily ET | Land cover class | | | |
|---|---|---|---|---|---|---|
| | | | Maize | Non-vegetation | Vegetables | Woods |
| 2012/06/29 | EC aggregation | 6.90 | 7.95 | 2.80 | 6.92 | 6.53 |
| | Modified P-M | 5.26 | 5.88 | 2.58 | 4.88 | 5.12 |
| 2012/06/30 | EC aggregation | 6.30 | 7.30 | 1.56 | 6.26 | 6.43 |
| | Modified P-M | 5.18 | 5.78 | 2.55 | 4.80 | 5.11 |



**Table 6** Comparison of total ET for each land cover class between remotely sensed ET product and EC aggregated ET (unit:

$m^3\ d^{-1}$, %)

| Land cover class | 2012/06/29 | | | 2012/06/30 | | |
|---|---|---|---|---|---|---|
| | EC aggregated | P-M Method | Under-estimation | EC aggregated | P-M Method | Under-estimation |
| Maize | 57589 | 42619 | 25.99 | 52881 | 41897 | 20.77 |
| Vegetables | 2349 | 1658 | 29.42 | 2125 | 1631 | 23.25 |
| Woods | 7431 | 5820 | 21.68 | 7317 | 5816 | 20.51 |
| Non-vegetation | 4304 | 3960 | 7.98 | 2398 | 3917 | -63.36 |
| All classes | 71673 | 54057 | 24.58 | 64721 | 53261 | 17.71 |

remark: "Underestimation" denotes the ratio of the difference between remotely sensed ET products and EC aggregated ET to the EC

aggregated ET value in percent, unit: [%].





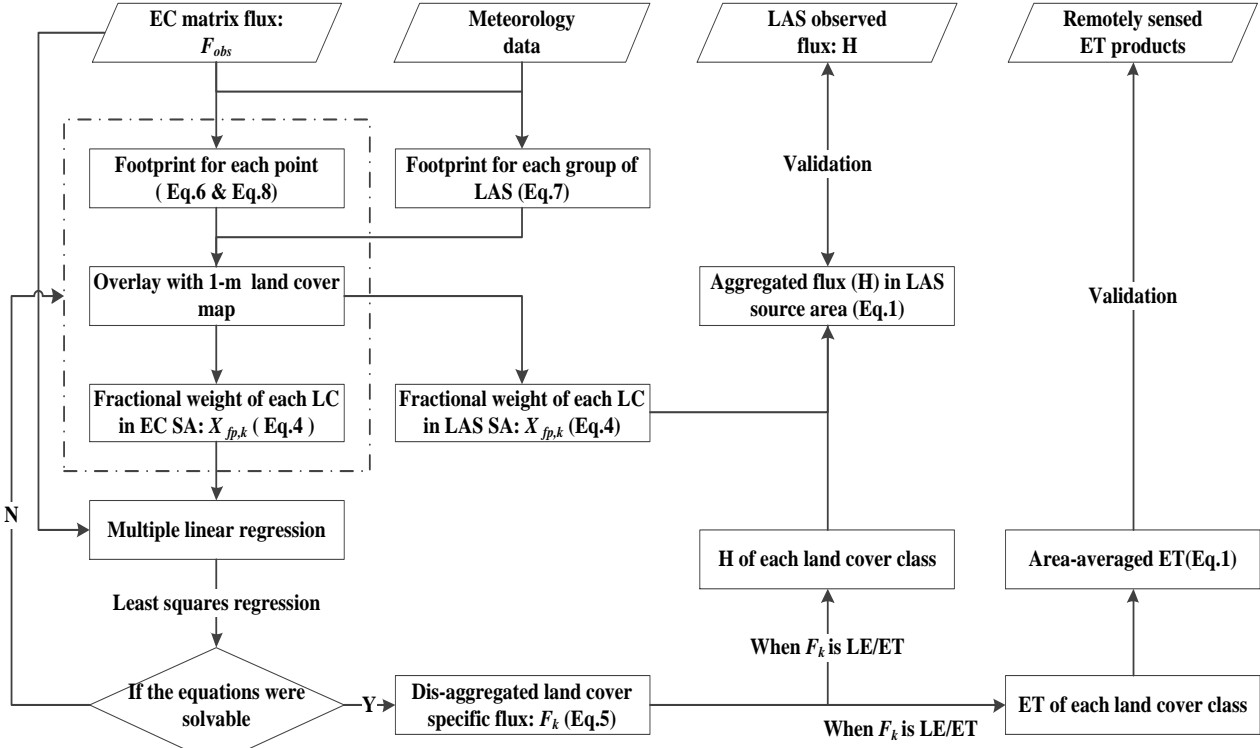

**Fig. 1** Schematic illustration of processing steps; LC = land cover class; SA = source area; H = sensible heat flux; LE = latent heat flux; ET = evapotranspiration





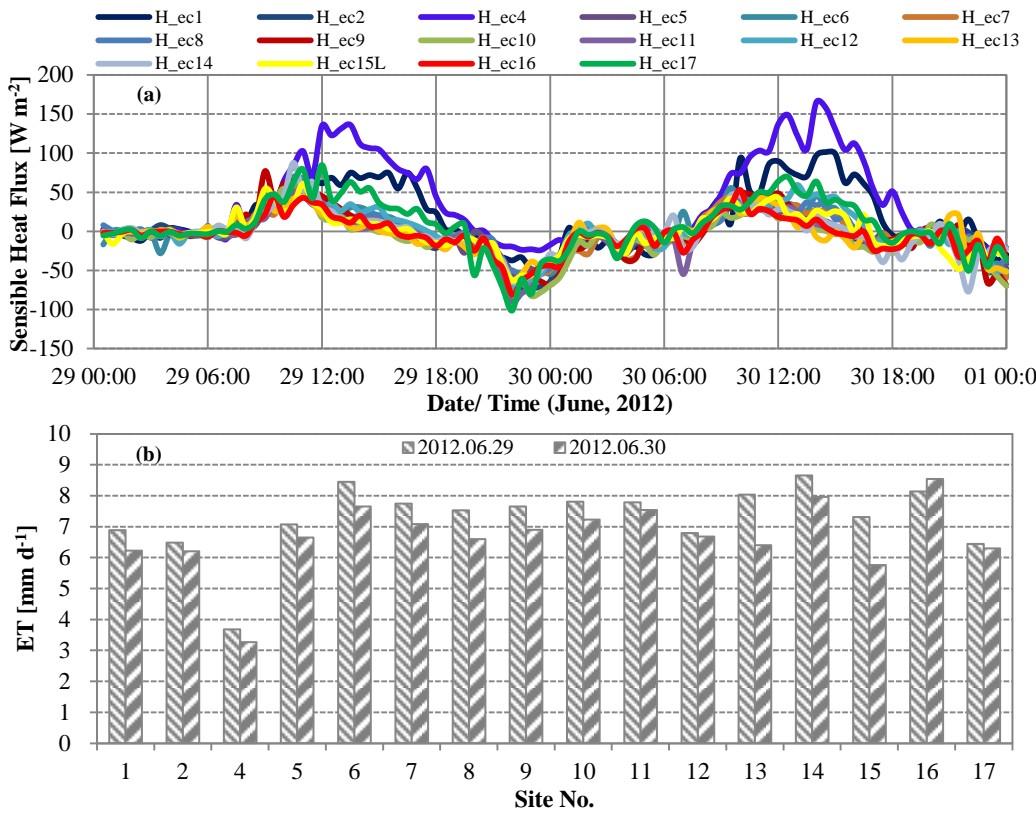

**Fig. 2** (a) Diurnal cycle of the sensible heat fluxes and (b) daily ET between different sites on 29 and 30 June, 2012





**Fig. 3** The land cover map of the kernel experiment area of HiWATER 2012. The small red circles represent the 90 % flux

contribution source area of EC sites, and the large blue circles covering different land cover classes indicate the source area

of LAS sites on 29 June 2012



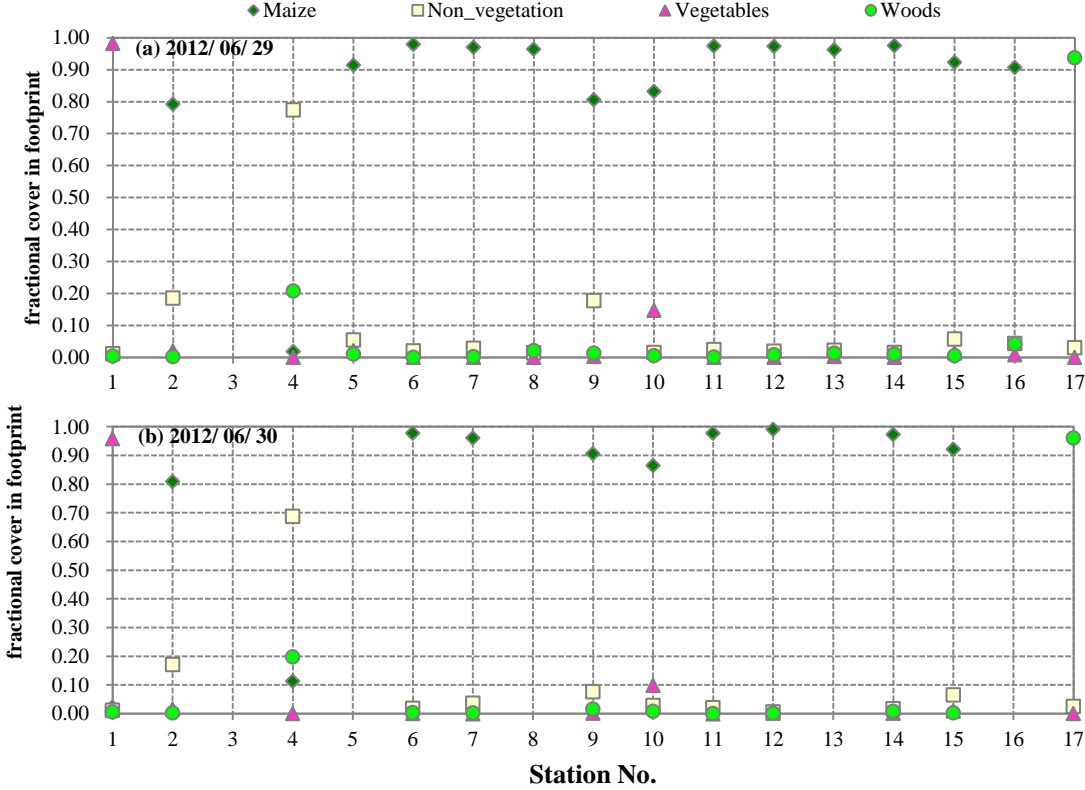

**Fig. 4** The fractional weight of each land cover classes in the daily averaged flux footprint of each EC flux measurements on

29 and 30 June 2012





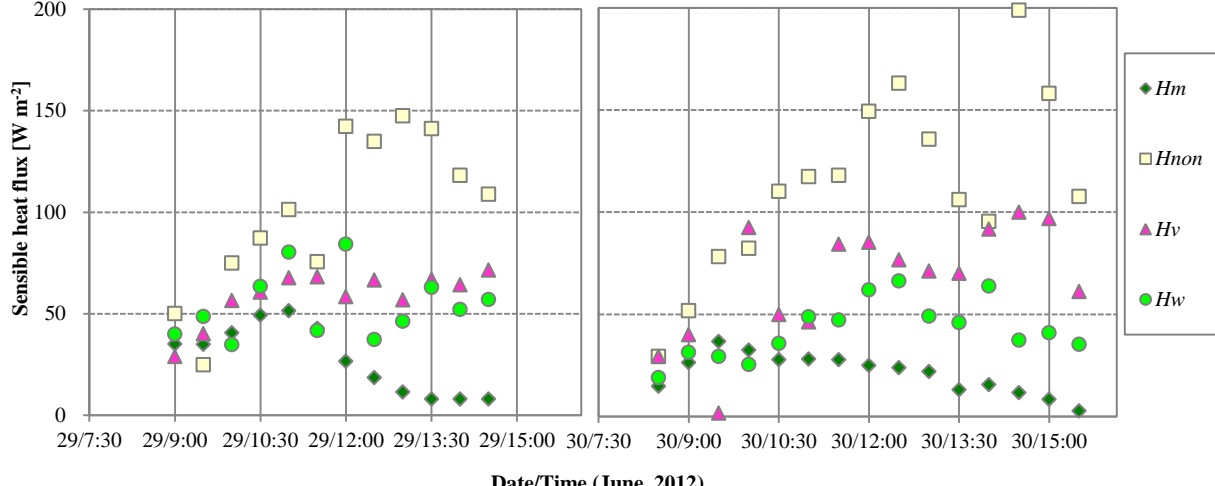

**Fig. 5** The diurnal cycle of the sensible heat flux for each land cover classes on 29 and 30 June 2012 (*Hm*,*Hnon*,*Hv*,*Hw*
indicate the sensible heat flux for maize, non-vegetation, vegetable and woods types, respectively )




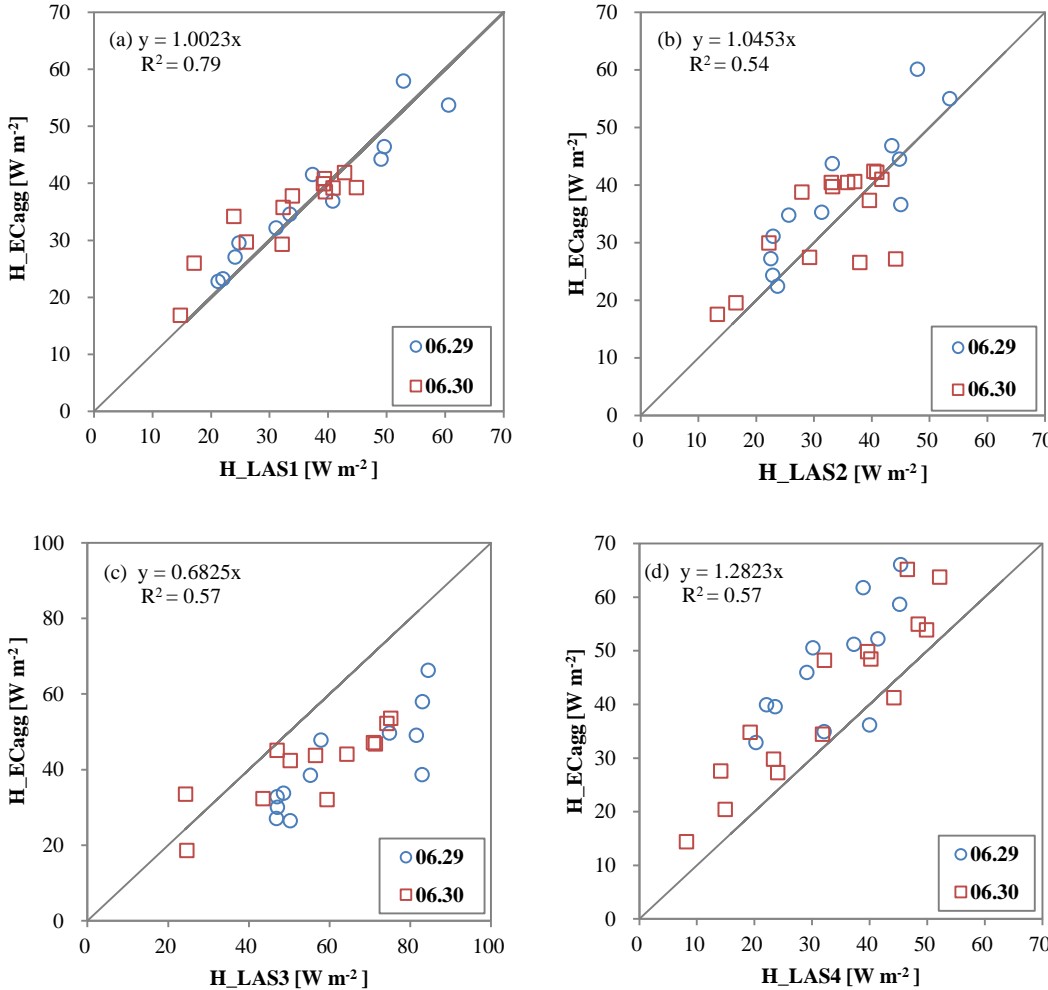

**Fig. 6** The comparison between LAS observed fluxes (X axis) and EC aggregated fluxes (Y axis)





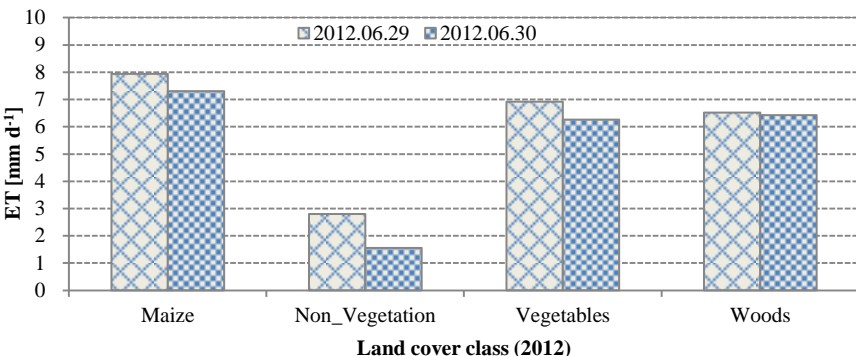

**Fig. 7** The dis-aggregated daily ET of each land covers in the kernel experimental area of HiWATER on 29 and 30 June 2012



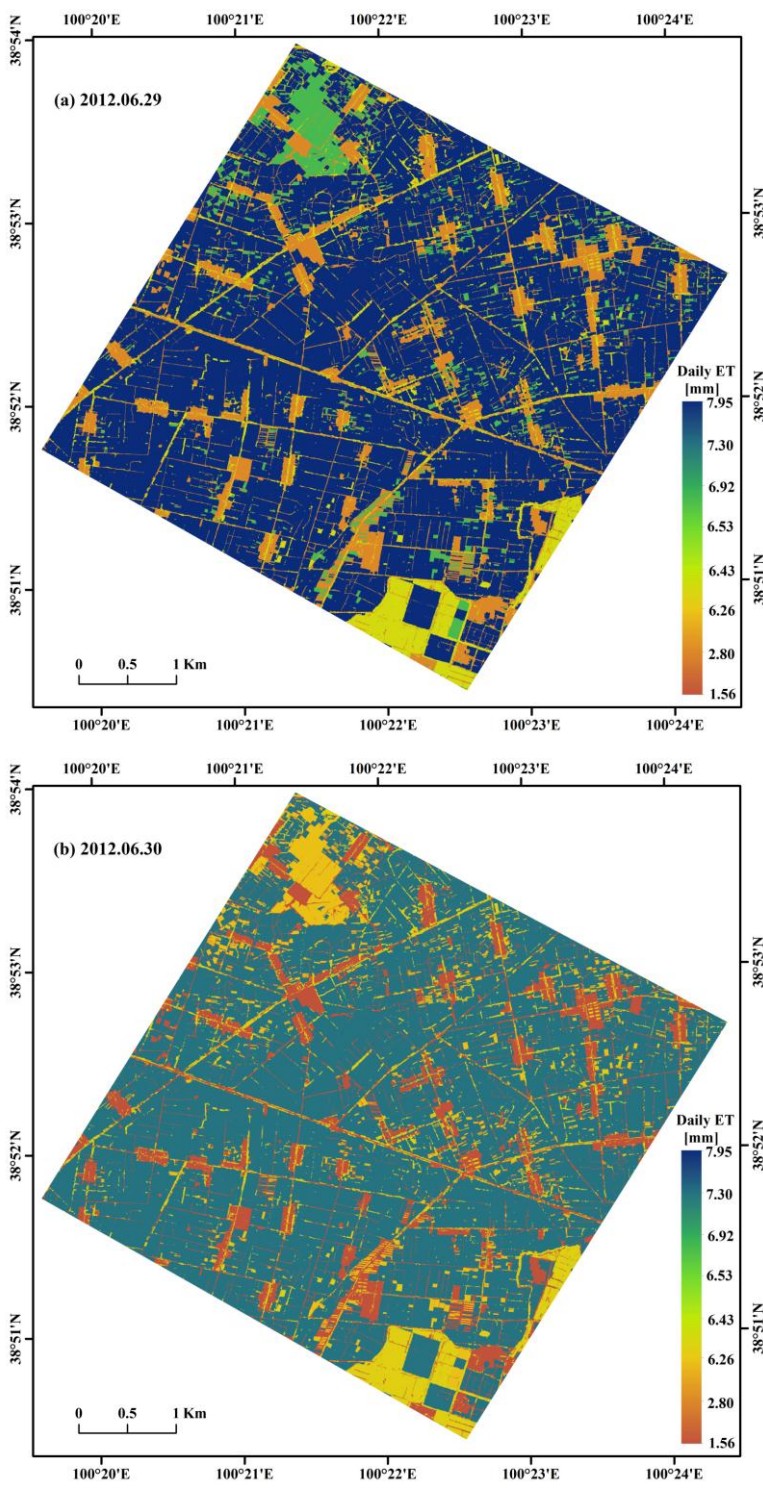

**Fig. 8** Spatial distribution of averaged daily ET in the kernel experimental area of HiWATER