# Peer review of "Area-averaged evapotranspiration over a heterogeneous land surface: Aggregation of multi-point EC flux measurements with high-resolution land-cover map and footprint analysis"

_Hydrology and Earth System Sciences, 2016_

## Referee Comment (RC1) · Anonymous Referee #1 · 16 Dec 2016

Accurate estimation of regional evapotranspiration (ET) remains a change in hydrology community. This paper took advantages of the HiWATER experiments to develop an aggregation scheme for accurate estimation of regional ET over an oasis area. In combination with a footprint model and a linear regression, the authors compared the ET aggregated from multi-site eddy covariance measures with that estimated from four large aperture scintillometers. The study addressed an important issue, with quality-controlled data obtained from the well-designed experiments. There are several concerns should be clarified before its acceptance for publication. First, in Introduction section the authors argued that existing integration schemes often assume local flux

measures are representative of an individual surface cover and thus result in errors. However, the paper did not explicitly address the issue with the data they used. The reviewer would like to see clearly to what extent and under what conditions the assumption may produce the errors. Second, while EC and LAS measures are valuable for large-scale ET estimation, they have measurement errors, either systematic or random. The errors are mixed or propagated into aggregated ET values. Are the measurement errors larger or smaller than the ET differences due to spatial heterogeneity? A careful analysis between EC, LAS and spatial heterogeneity with the matrix flux data would provide valuable insights into the issue which is puzzling for many years. Third, section 4.4 seems not closely relevant to the aggregation topics addressed in the paper. Neither the ET estimates can be validated over the study area as a whole. It is better to remove it from the text.

Specific: Abstract: Page 1 line 20-23: it does not provide any new for audience. 1.Introduction Page 2 line 13: earth -> Earth Page 3 line 2-3: remove one of "remote sensing" Page 3 line 8: there is -> there may be Page 4 line 9: a nice statement on representativeness of flux measures over individual surface covers. However, the present version failed to explicitly address the assumption in Results and Discussion. Page 5 line 1-2: "disaggregation approach has not been fully investigated" does not absolutely mean it deserves investigation. Please state it more clearly. 2. Study sites and data Page 6 line 14: two days only? Are they representative or enough to get conclusions that are general? Line 16: Please state the last time the irrigation done. Page 7 line 16: It would be better to use local time. Otherwise, explicitly state the time difference to Beijing time. There are many places throughout the paper with mixture use of "remotely sensed", "remote-sensing", "remote-sensing based", "satellite-based", etc. 3. Methodology Page 9 line 8: add a reference here for footprint model. Page 9 line 11: remove "The". There are many places with misuse of "the" or "a/an" Page 11 line 10: what the mean of "footprint climatology function"? Page 12 line 23: framework -> data processing flow. 4. Results and discussion Page 12 line 17-19: remove the paragraph. It is useless here. Please use W/m2 instead of mm/day throughout the paper. Page 13

line 13: how about other periods? Do the differences change with different periods? Page 16 line 16-page 17 line 10: remove the text that describes regional ET over the study area as a whole. It provides no support for the scientific issues addressed.

---

## Referee Comment (RC2) · Anonymous Referee #2 · 29 Dec 2016

This paper describes aggregation of multi-point EC flux measurements using an excellent set of EC measurement data and airborne remote-sensing data. The work presented in this paper is valuable, as the foot-print of ground measurement is always the issue on validating satellite remote-sensing ET estimations. By reviewing the results presented in the paper, it seems that the authors have derived the results using appropriately quality-controlled EC measurement data.

Major Comments: While the authors' works are valuable, the size of paper is too large, with containing less-important information. I recommend authors to drop entire section

4.4 and the related descriptions available in other sections (section 3.3 etc.). I could not find a purpose of Fig 8, and comparisons with "remotely sensed ET data" (Table 5 and 6). Also, including the comparison with remotely-sensed ET data in this paper might derive another problem on reviewing process because the procedure adopted in the paper is not well described in the paper, and the applied method may not be appropriate.

Minor Comments: (1) Page 8 Line 10; Authors manually revised land cover map using high-resolution CCD images and Google Earth imagery. Do those images applicable for year 2012? (2) By reviewing the results, the EC data used in the paper seems to be reliable. However, it is better to describe in the paper some more about the measurement accuracy of their EC data, for example, about the energy closure error. (3) It is authors' preference and authors do not need to change, but I might recommend changing Fig 2-b (ET) from bar-graph with mm/d, to line graph with W/m2 like Fig 2-a, so readers can understand the energy balance condition of the sites by directly comparing Fig 2-a and 2-b. (4) It is authors' preference and authors do not need to change, but Fig.4 can be expressed not as figure but as table. (5) In Fig 6, I recommend authors to show the character "a" "b" "c" and "d" in the figure, because authors are referring the figure such as "fig 6c" in the text.

---

## Referee Comment (RC3) · Anonymous Referee #3 · 31 Dec 2016

General comments: Due to surface heterogeneity, it has always been a challenge to validate remote sensing heat fluxes with field measurements. This paper proposed an aggregation scheme to derive area-averaged heat fluxes from multi-point EC measurements, combining multivariate regression and footprint analysis. It provides a new method to compare fluxes measured by EC with those either measured by LAS or modelled using satellite data, which was evaluated with a complete and valuable dataset-HiWATER. However, there are still several issues to be considered. To address the scientific problem in this paper, 30-min flux might be sufficient, given the uncertainty in gap-filling methods (rainfall, fog etc.). Daily regional ET (section 4.4) does not help a

lot here. Instead, it might be more necessary to clarify data quality and uncertainty of the EC and LAS measurements. Besides, P-M estimated ET could be removed.

Specific comments: Page 5 Line 24 ' following Fig .3' Since it is the first figure appearing in this article, it's better to change the number from 3 to 1. Page 6 Line 15 ' EC data from 16 towers...' According to section 4.2, in addition to site 3, sites 5/8/13/16 were also not used. It is better to use a consistent dataset through the paper. Page 6 Line 16 ' no irrigation' And how was the weather during this period? Page 6 Line 22 ' coordinate rotation' Why not use Planar Fit? Page 6 Line 13 'MOST' There are different solutions. Add either corresponding equations or references here. And how were roughness height and zero-plane displacement estimated? Page 6 Line 16 ' daytime...'. It's a bit confusing. Local time is better. For data quality control, what is the threshold value of signal strength? Section 2.2.2 This section could be abbreviated if the preliminary land cover has already been done by Liu and Bo(2015). Page 8 Line 11, specify the date of the google earth image. Page 8 Line 16-20, it might not be necessary to compare with PM- ET. The principle of that is the same as the comparison with LAS in terms of flux aggregation and there might uncertainty in PM-ET. Section 4.3 Page 15 & Page 16. It's better to look into the details to figure out the factors contributing to the bias between EC and LAS, instead of just mentioning 'heterogeneous distribution of surface covers'. Section 4.4 I didn't see the difference between Table 5 and 6 in terms of addressing the problem despite their different units.

---

## Referee Comment (RC4) · T. Foken (Referee) · 17 Jan 2017

Reading the manuscript, I found that the concept of the experimental design and the data analysis is very similar to the experiment LITFASS-2003, which was published in BAMS (Mengelkamp et al., 2006) and in a special issue of Boundary-Layer Meteorology (2006, vol. 121, issue 1). Some of these papers are quoted, but papers published later are missing (Foken et al., 2006; Foken et al., 2010; Charuchittipan et al., 2014).

Several parts in the paper are unclear, or information is missing that would enable the paper to be followed accurately:

[Figure]

The area of investigation was very much dominated by maize fields. Only three stations had another dominant land cover (stations 1, 4, and 17). This is a significant limitation for the stated aim of the paper to determine area-averaged fluxes over a heterogeneous area. For the LITFASS-2003 experiment (and other experiments given as references), different land cover types were much better distributed. This deficit should be discussed.

The function of the LAS in the aggregation schema was not clear. I could not find a reason for the use of such data. In LITFASS-2003, LAS systems were also used with a specific function: It was assumed that LAS can also measure the fluxes of larger turbulence or circulation structures and that this is not affected by the non-closure of the energy balance (Foken, 2008). This information was used to discuss the unclosed energy balance of the flux measurements and to correct this. The problem of the unclosed energy balance is not mentioned in the whole paper, but it is a standard for the analysis of surface flux measurements (Foken et al., 2012).

Any information is missing as to why the footprint model by Kormann and Meixner (2001) was used in your study. Perhaps the textbook by Leclerc and Foken (2014) would give you the relevant information. Questionable is the exact location of the small non-maize-covered areas in the footprint of the EC and LAS measurements. A discussion of the accuracy of the footprint analysis combined with the accuracy of the EC and LAS measurements is urgently necessary.

The applied multiple-linear regression analysis needs more information. Did you aggregate the fluxes according to the land-cover type in different effect levels of the footprint? Compare your method with the methods presented by Leclerc and Foken (2014).

What is meant by "Remotely sensed ET products"? If I understood the paper correctly, only the land-cover type was determined by satellite measurements, but, as seems probable, did these also include the net radiation for use in the Penman-Monteith equation? But this would then be difficult for the heterogeneous land cover. It is impossible

to discuss the underestimation of the fluxes by the Penman-Monteith equation without knowing the parameterizations used in this equation. E.g., the atmospheric resistance and the stomata resistance are extremely variable and should be included in any discussion.

Please also show in Fig. 2 the daily cycle of the evapotranspiration and not only the daily sum. This is necessary to indicate the energy exchange of the different sites, possible oasis effects, and the Bowen ratio. The latter may be a good indicator which to classify the sites.

Undoubtedly the authors have an interesting data set with a significant scientific potential. Such a data set should be published with a good scientific concept. Besides some deficits in the experimental design, the concept of area-averaged fluxes may be such a concept. But the paper needs significant improvements according to the points given above. Therefore I recommend major revisions.

Minor remarks:

The numbering of the figures is confusing. Figure 3 should be renamed as Fig. 1.

Table 1: The instrumentation (sonic anemometer, gas analyzer) is missing.

Table 2: Do not mix LAS type and LAS producer, please give both for all sites.

p. 6, line 21: What do the flags mean?

p. 6, line 23: Why did you use 2D-rotation and not planar fit? Was the terrain absolutely even?

p. 7, line 13: L can be easily misinterpreted as the Obukhov length in a micrometeorological paper.

Fig. 4 and 5: Why did you use different names or land cover types in both figures?

Fig. 6: Probably y has a lower accuracy than given in the figure!

p. 16, line 11: The reference should probably be Fig. 3!

p. 17, line 12: This is trivial; when maize dominates the land cover it is normal that maize also dominates the ET.

Table 6: Give the units in the columns.

p. 19, line 16-25: Such a paper needs a well-written conclusion chapter and not only ten not very significant lines.

p. 22, line 13: Many authors are missing.

p. 22, line 18: Print $CO_2$.

References

Charuchittipan, D., Babel, W., Mauder, M., Leps, J.-P., and Foken, T.: Extension of the averaging time of the eddy-covariance measurement and its effect on the energy balance closure Boundary-Layer Meteorol., 152, 303-327, 10.1007/s10546-014-9922-6, 2014.

Foken, T., Wimmer, F., Mauder, M., Thomas, C., and Liebethal, C.: Some aspects of the energy balance closure problem, Atmos. Chem. Phys., 6, 4395-4402, 2006.

Foken, T.: The energy balance closure problem – An overview, Ecolog. Appl., 18, 1351-1367, 10.1890/06-0922.1, 2008.

Foken, T., Mauder, M., Liebethal, C., Wimmer, F., Beyrich, F., Leps, J.-P., Raasch, S., DeBruin, H. A. R., Meijninger, W. M. L., and Bange, J.: Energy balance closure for the LITFASS-2003 experiment, Theor. Appl. Climat., 101, 149-160, DOI 10.1007/s00704-009-0216-8, 2010.

Foken, T., Leuning, R., Oncley, S. P., Mauder, M., and Aubinet, M.: Corrections and data quality in: Eddy Covariance: A Practical Guide to Measurement and Data Analysis, edited by: Aubinet, M., Vesala, T., and Papale, D., Springer, Dordrecht, Heidelberg,

London, New York, 85-131, 2012.

Kormann, R., and Meixner, F. X.: An analytical footprint model for non-neutral stratification, Boundary-Layer Meteorol., 99, 207-224, 2001.

Leclerc, M. Y., and Foken, T.: Footprints in Micrometeorology and Ecology, Springer, Heidelberg, New York, Dordrecht, London, XIX, 239 pp., 2014.

Mengelkamp, H.-T., Beyrich, F., Heinemann, G., Ament, F., Bange, J., Berger, F. H., Bösenberg, J., Foken, T., Hennemuth, B., Heret, C., Huneke, S., Johnsen, K.-P., Kerschgens, M., Kohsiek, W., Leps, J.-P., Liebethal, C., Lohse, H., Mauder, M., Meijninger, W. M. L., Raasch, S., Simmer, C., Spieß, T., Tittebrand, A., Uhlenbrook, S., and Zittel, P.: Evaporation over a heterogeneous land surface: The EVA_GRIPS project, Bull. Amer. Meteorol. Soc., 87, 775-786, 2006.

---

## Author Comment (AC1) · 19 Jan 2017

Dear Referee #1:

We appreciate very much for your valuable comments and suggestions on our manuscript. According to your comments, and those from Referee #2 and Referee #3, we have carefully revised all sections of the paper (revisions and corrections are marked in red). Detailed response to your worthwhile comments and suggestions are as follows:

Main comments: 1. In Introduction section the authors argued that existing integration

schemes often assume local flux measures area representative of an individual surface cover and thus result in errors. However, the paper did not explicitly address the issue with the data they used. The reviewer would like to see clearly to what extent and under what conditions the assumption may produce the error.

Response: Thanks for your valuable comments. The three aggregation methods, particularly the simple arithmetic and/or the area-weighted method used before, are based on individual surface types, without high resolution land-use classification and fine footprint analysis. We have revised the relevant section in "Introduction" and believe a better result would be achieved based on present integration method.

2. While EC and LAS measures are valuable for large-scale ET estimation, they have measurement errors, either systematic or random. The errors are mixed or propagated into aggregated ET values. Are the measurement errors large or smaller than the ET differences due to spatial heterogeneity? A careful analysis between EC, LAS and spatial heterogeneity with the matrix flux data would provide valuable insights into the issue which is puzzling for many years.

Response: Yes, both EC and LAS have measurement errors, either systematic or random. For the ECs used in our data analysis (the HiWATER) we have tried to reduce the systematic errors to a minimum with a pre-observation intercomparison (Xu et al., 2013 JGR) and careful maintenance during the observation. The random errors were also analyzed via a separate research (Wang et al., 2015, IEEE GRSL), which can be minimized in an ensemble average. As for the LASs used in HiWATER, major errors are from their data processing processes, for instance, the Bowen-ratio correction problem particularly for observations over Oasis. We have also tried our best to minimize them mainly through intercomparisons with fluxes from EC. All these uncertainties are minor when we compared with the spatial heterogeneity of our study area, especially, the large differences among the four kinds of land-use. We have added some descriptions on data quality and uncertainty of the EC and LAS measurements in our study in Section 2.2.1.

3. Section 4.4 seems not closely relevant to the aggregation topics addressed in the paper. Neither the ET estimates can be validated over the study area as a whole. It is better to remove it from the text.

Response: We are sorry for our unclear statements in section 4.4. We have removed irrelevant information including the comparison with P-M ET products. The statements in the entire section have been re-written.

Specific comments:

Abstract: Page 1 line 20-23: it does not provide any new for audience.

Response: Accepted: Line 20-23 are deleted.

1. Introduction Page 2 line 13: earth - > Earth Page 3 line 2-3: remove one of "remote sensing" Page 3 line 8: there is - > there may be

Response: Accepted.

Page 4 line 9: a nice statement on representativeness of flux measures over individual surface covers. However, the present version failed to explicitly address the assumption in Results and Discussion.

Response: Thanks for your comments. We have added some statements in Section 4, Results and Discussion, to address the assumption explicitly.

Page 5 line 1-2: "disaggregation approach has not been fully investigated" does not absolutely mean it deserve investigation. Please state it more clearly.

Response: We have improved the statements in this section more clearly.

2. Study sites and data Page 6 line 14: two days only? Are they representative or enough to get conclusions that are general?

Response: The two clear days we selected for analysis, 29 to 30 June 2012, are typical (due to the weather, surface status, and extended observations such as aircraft remote

sensing) and representative for the general conclusions we got. Actually, we have applied the same flux aggregation method for other periods, such as the 6 days from 9 July to 14 July 2012. Figure 1 (attached below, similarly hereinafter) shows the daily ET for four land covers derived. Figure 2 describes the area-averaged daily ET over the study area. All the areal ET and its disaggregation to individual land types are similar to those of the two clear days analyzed.

Page 6 line 16: Please state the last time the irrigation done.

Response: We have added the last irrigation time in the revised manuscript.

Page 7 line 16: It would be better to use local time. Otherwise, explicitly state the time difference to Beijing time.

Response: The time difference between local time and Beijing Standard Time is approximately +1 h 18 min. This has been added explicitly in the revision.

There are many places throughout the paper with mixture use of "remotely sensed", "remote-sensing", "remote-sensing based", "satellite-based", etc.

Response: We have unified the use of "remote sensing" throughout the revised paper.

3. Methodology

Page 9 line 8: add a reference here for footprint model.

Response: A reference has been added: Leclerc, M. Y., and Foken, T.: Footprints in Micrometeorology and Ecology, Springer, Heidelberg, New York, Dordrecht, London, XIX, 239 pp., 2014

Page 9 line 11: remove "The". There are many places with misuse of "the" or "a/an"

Response: Accepted. Other places with misuse of "the" or "a/an" have also been checked.

Page 11 line 10: what the mean of "footprint climatology function"?

Response: We have revised the "footprint climatology function" as "the weighted footprint climatology". Its meaning is clearly shown with Eq. (8).

Page12 line 23: framework -> data processing flow.

Response: Accepted.

4. Results and discussion Page 12 line 17-19: remove the paragraph. It is useless here.

Response: Accepted.

Please use W/m2 instead of mm/day throughout the paper.

Response: We have unified the flux unit from mm/d to W/m2 in Section 4.1. However, in discussing the daily ET in Section 4.4 we still use mm/d as in usual applications.

Page 13 line 13: how about other period? Do the differences change with different periods?

Response: The differences among all maize sites change slightly during the maize grown period. For example, Fig. 3 shows the daily variation of the three major fluxes from 9 July to 14 July 2012. The standard deviations of all maize sites are also shown. All features are about the same as the two days analyzed. We have added some relevant statements in the revised paper.

Page 16 line 16-page 17 line 10: remove the text that describes regional ET over the study area as a whole. It provides no support for the scientific issues addressed.

Response: The major objective of this study is to refine an aggregation method for area averaged fluxes based on our unique, comprehensive dataset of the HiWATER. The results are also useful for the water balance study extended to the whole Heihe River basin. So the results of regional ET over the study area are still kept in Section 4.4. Of course, some irrelevant parts are deleted according to the comments of yours and other two referees.

Thank you very much again for your valuable comments and suggestions on our manuscript. The revised manuscript is attached as supplement.

Sincerely yours, Feinan Xu Email: xufeinan@lzb.ac.cn

PS. After revising our manuscript and finishing the above responses to you, yesterday, we received the comments from Prof. Thomas Foken (as Referee #4). Some important revisions would be needed based on his comments. A new version might be uploaded within two weeks.

Corresponding author: Weizhen Wang, weizhen@lzb.ac.cn; Northwest Institute of Eco-Environment and Resources, Chinese Academy of Sciences, 320 Donggang west road, Lanzhou, Gansu, 730000, China.

Please also note the supplement to this comment:
http://www.hydrol-earth-syst-sci-discuss.net/hess-2016-602/hess-2016-602-AC1-supplement.pdf

[Figure]

**Fig. 1.** EC dis-aggregated daily ET for each land covers from 9 to 14 July 2012

Fig. 2. Area-averaged daily ET over the study area

Fig. 3. Diurnal cycle of the mean net radiation (a), sensible (b) and latent (c) heat fluxes for 13 maize field sites, the errors bars are the standard deviation.

**Supplement:**

[revised manuscript text omitted]

For the ECs used in our data analysis, the systematic errors had been tried to reduce to a minimum with a pre-observation intercomparison (Xu et al., 2013) and careful maintenance during the observation. The random errors were also analyzed by a separate research (Wang et al., 2015), which can be minimized in an ensemble average. As for the LASs, the errors from data processing processes (e.g. the Bowen-ratio correction problem) were also tried to minimize as much as possible through intercomparisons with fluxes from EC.

For the ECs, the energy balance closure ratio (EBR) was also assessed (Xu et al., 2017). The ratio

of the turbulent heat flux (the sensible and latent heat flux) to available energy for 16 EC sites over vegetated surfaces was 87 %, and the EBR for site 4 was 0.84.

**2.2.2 Collection and processing of remote-sensing data products**

A land cover classification map with 1-m spatial resolution was collected. The map was derived from 
[revised manuscript text omitted]
 values of these input parameters can be derived from measurements generally available from EC flux towers. The daily flux contribution area of the EC flux measurements was calculated by Eq. (8), which provides approximately 90 % of the total source area that contributes to the measured fluxes. Every 30-min flux source area of the LAS sites was estimated via Eq. (7), and the 90 % half-hourly footprint contours of LAS measurements were used. The normalized daily footprint climatology of ECs and half-hourly footprint estimates of LASs were individually overlaid

with 1-m land cover map to determine the footprint-weighted contribution of each land cover classes to the measured flux from EC and LAS systems.

**3.3 Data processing flow of the determination of area-averaged fluxes**

The overall data processing flow for determining the area-averaged evapotranspiration over a heterogeneous land surface mainly includes three aspects (Fig. 2).

Firstly, the spatial representativeness of the 16 EC flux towers within the $5 \times 5$ km$^2$ experiment area was quantitatively assessed by overlaying in-site flux footprint climatology with 1-m land cover map. Detailed analyses on this aspect are going to be presented in the following section.

The second aspect was to evaluate the reliability of the established flux aggregation schemes. The land cover specific flux was firstly dis-aggregated from multiple EC flux measurements by performing a multiple linear regression analysis (Eq. 5). The EC dis-aggregated fluxes of each land cover classes were then aggregated again to obtain area-averaged fluxes, according to the fractional weight of each land cover class in the LAS footprint (Eq. 4). Finally, the aggregated fluxes were compared with LAS observations.

At last, the area-averaged evapotranspiration over a heterogeneous land surface was estimated from multi-point EC flux measurements with the developed flux integration schemes, based on footprint analysis and high resolution land cover map.

**4 Results and Discussion**

**4.1 The characteristics of the surface heat and water vapor fluxes**

Figure 3 depicts the diurnal cycle of the sensible (Fig. 3a) and latent (Fig. 3b) heat fluxes at different sites on two clear days. Both of the two figures reveal significant differences in the magnitude of the sensible- and latent heat fluxes between different surface types during the growing season.

The sensible heat flux over residential area reached a maximum of about 150 W m$^{-2}$ at afternoon

and was higher than that over the vegetated surfaces (H_ec4, Fig. 3a), while the latent heat flux was smaller compared with other sites, with maximum value of less than 300 W m$^{-2}$ due to a certain fraction of sealed land surfaces (LE_ec4, Fig. 3b).

Over the vegetated surfaces (orchard, vegetable, maize), the sensible heat flux was nearly less than 100 W m$^{-2}$ because of well-irrigated cropland (Fig. 3a). The sensible heat flux over the three types of vegetation was also significantly different, and there was also a difference in sensible heat fluxes among maize sites. The mean value of the standard deviation (SD) of H for 13 maize sites was about 8.4 W m$^{-2}$ (Fig. 4a).

Deviations in latent heat fluxes over different vegetation types were also found (Fig. 3b, Fig. 4b). The maize fields performed highly latent heat fluxes and lower sensible heat fluxes than the other two vegetated surfaces. One of the possible reasons is that both of the orchard area and the vegetable field are relatively sparse compared with the maize cropland.

The mean value of the SD of LE for all maize sites was approximately 43.3 W m$^{-2}$. The result showed that the latent heat flux over maize cropland exhibited larger SD than the sensible heat flux, and it also indicated the LE differed between sites for same underlying surface (Fig. 4). This can be partly explained by the discrepancy in plant physiology and vegetation growing stage.

The preliminary results indicated that the variability and difference in the surface energy fluxes between the HiWATER tower flux sites was significant during the crop growth period. The differences in sensible and latent heat fluxes between maize field sites also could be noticed.

**4.2 Analysis of the representativeness of the multi-point EC flux measurements**

To further understand the variability of surface energy fluxes between different sites in a heterogeneous landscape, the footprint analyses for representativeness of EC sites were applied by overlaying flux footprint with high resolution land-cover map (Fig. 1). The fraction of land cover classes present in the daily-averaged footprint of each EC measurements is given in Fig. 5. Given the

source area (90 % flux contribution) of the 4 ECs (sites 5, 8, 13 and 16) on 30 June 2012 exceeded the extent of land cover map, the spatial representative of the 4 EC sites were not shown in Fig. 5b.

Due to the variations in the observation height, atmospheric stability, wind direction and wind speed, the exact shape and size of the EC source area were distinctly different (Fig. 1). For each EC flux measurements, there was more than one type of land cover in its footprint. The contribution of each land cover classes to the total measured flux for EC sites was changed with the varying source area (Fig. 5).

The dominated surface types in the source area were vegetable and orchard at sites 1 and 17, respectively. For site 4, however, there were mainly three types of land cover within its footprint, namely non-vegetation, maize and woods type. The fractional weight of the non-vegetation type and maize field in the footprint greatly varied, while the proportion of woods was almost changeless.

At maize field sites, the relative contribution of maize field to the EC measured flux was approximately more than 0.9, except for sites 2, 9 and 10. At site 2, the percentage of non-vegetation type in the footprint was almost 0.18. For site 9, the rate of maize and non-vegetation type present in footprint significantly varied. The contribution of vegetable type to the flux measurements at site 10 ranged from 0.15 to 0.1.

The above analysis shows that the tower flux measurements at the field scale are generally representative of multiple surface types. The result indicates that the latent and sensible heat fluxes measured by EC systems are representative of the averaged fluxes, which are weighted the upwind surface flux emanating from individual land cover classes with flux footprint. In general, it may be problematic to validate the model estimated fluxes by direct comparison with tower-based flux measurements over a heterogeneous land surface. Thus, the extension of multiple flux measurements to pixel/grid scale surface fluxes is urgently needed.

**4.3 Evaluation of the EC aggregated fluxes**

The determination of area-averaged fluxes from point measurements is usually not straightforward,

especially for heterogeneous land surfaces. Based on multi-point EC flux measurements and accurate 1-m land cover map, a flux aggregation method was established to estimate averaged surface fluxes with footprint analysis and multivariate regression. Fig. 1 shows that all types of land covers present in the LAS flux footprint, so the LAS measurements can be taken as reference to assess the feasibility of the developed integration schemes.

At first, the sensible heat flux for each land cover was dis-aggregated from the EC observed component fluxes in a heterogeneous footprint with multiple linear regression method. The diurnal cycle of the EC dis-aggregated sensible heat fluxes for each land cover types is highly significant (Fig. 6). During the crop growth stage, the sensible heat flux over maize field showed a minimum value in the afternoon, while the sensible heat fluxes for non-vegetation type at daytime exhibited a maximum of about 200 W m$^{-2}$. The sensible heat fluxes for vegetable and woods types lied between them.

Then, the sensible heat flux representative for the LAS source area was aggregated by multiplying the EC dis-aggregated fluxes for the four land-cover classes by their relative fraction in the LAS source area. Fig. 7 illustrates a scatterplot of 30-min averaged sensible heat fluxes estimated using the flux aggregation method (hereafter referred as H_ECagg) versus LAS measurements (H_LAS), as well as the linear regression parameters (including equations and R$^2$). The different statistics between LAS observed fluxes and EC aggregated results are listed in Table 3.

For LAS 1 (see Fig. 7a and Table 3), a good agreement is found between EC aggregated fluxes and LAS measurements, with high correlation coefficient and low RMSE value (R$^2$= 0.79, RMSE= 0.96 W m$^{-2}$). The scatter points in the graph are nearly close to the 1:1 line. The MBE and MAPE values were 4.25 W m$^{-2}$ and 9.93 %, respectively.

Compared with LAS 1, there was a little scatter between LAS measured fluxes and estimates from multiple EC flux observations for LAS 2, but yielding a small mean bias error (MBE = 2.31 W m$^{-2}$) (Fig. 7b, Table 3). RMSE and MAPE values between H_ECagg and H_LAS2 were much higher than that of LAS 1, with values of 6.91 W m$^{-2}$ and 16.39 %, respectively. Considering the heterogeneous

distribution of surface covers in the LAS source area, slight area of non-vegetation distributing in the center of LAS 2 path would be the primary factor attributing to the bias (blue circles in Fig. 1).

For LAS 3 (Fig. 7c, Table 3), there was a slightly weak relationship between sensible heat fluxes derived from the LAS measurements and flux aggregation method, with correlation coefficient ($R^2$) of 0.57 and RMSE, MAPE and MBE values of 17.63 W m$^{-2}$, 31.7 % and -18.01 W m$^{-2}$, respectively. The scatter points in Fig. 7c were overall below the 1:1 line. It indicated that the 30-min averaged H estimated from EC flux observations using the aggregation method were underestimated against LAS derived H (negative MBE). As shown in Fig. 1, there is more large area of residential areas randomly distributing in the center of LAS 3 path than other three LAS systems. This discrepancy is likely related to the heterogeneously distributed surface types.

In Fig. 7d, the area-averaged sensible heat fluxes obtained using the flux aggregation method were consistent with LAS measurements, with $R^2$ of 0.57 for LAS 4. In contrast with LAS 3, the scatter points in this graph were almost above the 1:1 line (overestimate of EC estimated H, MBE > 10 W m$^{-2}$). RMSE value of LAS 4 relatively decreased by 4.88 W m$^{-2}$, but MAPE value was up to 33.7 %. The red open squares in Fig. 7d are more close to the 1:1 line than the blue open circles. When southeast wind prevailed, the relative contribution of non-vegetation type to the LAS 4 measurements was about 0.2, and its value decreased to 0.08 when the main wind direction was northwest.

Through detailed analysis, the magnitude of divergences between the estimated and measured area-averaged surface fluxes is in large part concerned with the variation of corresponding LAS source area. Moreover, the contribution of non-vegetation type to the LAS observations, which accounted for a large proportion in the footprint of LASs (LAS 2, 3 and 4), would be one of the main factors contributing to the bias between the estimated results and LAS measurements. The reason is that the EC dis-aggregated sensible heat flux for non-vegetation type may not be representative for the flux emanating from sealed buildings and roads that are part of non-vegetation type. On the other hand, the complexity of the LAS footprints may lead to the inconsistency of EC-estimated and LAS-measured

averaged fluxes.

Overall, the above results demonstrate that, compared with the area-averaged fluxes measured by LAS systems, the area-averaged fluxes that are aggregated from multiple EC flux measurements using the established flux aggregation method are reliable. Therefore, the developed flux integration schemes in this study can be an effective way to estimate areal averaged fluxes.

**4.4 Estimation of area-averaged evapotranspiration**

The flux aggregation scheme, which was established and evaluated in Sect. 4.3, was adopted to determine the area-averaged ET over our study area with multi-point EC flux measurements and high resolution land-cover map. The EC dis-aggregated daily ET for all the land covers over two clear days was shown in Fig. 8. As can be seen, the daily ET values for maize field were highest during the crop growing season (7 mm – 8 mm). The value of daily ET was 6.4 mm for woods type, and it ranged from 6 mm to 7 mm for vegetable type. On the contrary, the daily ET for non-vegetation type varied largely, with values of 2.8 mm on 29 June and 1.5 mm on 30 June, respectively.

The daily ET maps at 1-m resolution were produced through the dis-aggregated daily ET for all land cover classes, combined with the 1-m land classification map. Fig. 9 depicts the spatial pattern of daily ET on 29 and 30 June 2012. It can be seen from the legend in figure, the daily ET ranged from 1.56 to 7.95 mm during the two days, and with higher values on 29 June (Fig. 9a) for all land cover classes than that on 30 June (Fig. 9b). The maize field performed highest ET value and distributed widely, whereas other three types of land cover randomly distributed across the whole study area with quite different ET values.

Table 4 lists the total ET for different land cover classes and their proportion of the total area ET. The total ET for our study area was almost 169626.77 m$^3$ on 29 June, while it was about 152948.41 m$^3$ on 30 June. The results demonstrated that the ratio of ET for maize field to the total area ET was in excess of 80 %. In addition, the total rate of ET for both woods and vegetables types was approximately

13 %, and the ET value for non-vegetation type accounted for 4.83 % of daily totals on the average.

Finally, the area-averaged daily ET over the kernel experiment area of HiWATER was estimated via Eq. (1), with values of approximately 7.01 mm on 29 June and 6.32 mm on 30 June 2012.

**5 Summary and conclusions**

On the basis of 1-m accurate land cover map and multi-point ground-based flux measurements datasets from 16 EC systems and 4 groups of LAS systems during the intensive observation period of HiWATER program, the area-averaged surface fluxes over a heterogeneous surface were determined by a flux aggregation method, which was established through the integration of footprint analysis and multiple regression. The estimated area-averaged fluxes were validated by the LAS measurements to assess the reliability of the integration method. Ultimately, the method that had been evaluated was used to estimate area-averaged ET over our study area.

First and foremost, analyses of the spatial representativeness of multiple EC flux towers were performed for the interpretation of the surface fluxes over different land surfaces. It is proved that the combination of footprint analysis and high-resolution land cover map can be a proper way to clarify the relationship between the tower-based flux observations over heterogeneous surfaces and individual land cover specific fluxes, and it is also the foundation for the establishment of flux aggregation scheme.

Secondly, based on good multi-scale (EC & LAS) flux datasets, precise flux footprints of flux towers and better land cover classification map, a flux aggregation scheme can be successfully established through the integration of footprint analysis and multivariate regression. In a heterogeneous study area, the surface flux emanating from individual land cover classes need to be firstly acquired by a flux integration method before deriving area-averaged fluxes. The results show that the developed flux aggregation method provides a unique opportunity to disentangle the heterogeneous land surface fluxes in their single components.

Then, the averaged surface fluxes estimated from the established flux aggregation method were

compared with the corresponding observed LAS values. The MAPE values of the LAS 1, LAS 2, LAS 3 and LAS 4 were 9.93 %, 16.39 %, 31.7 % and 33.7 %, respectively. The reasons for the divergences between EC aggregated estimates and LAS observations were investigated by a combination of remote sensing data and ground measurements, and the findings revealed that the extent of vegetation structure heterogeneity had a significant influence on the application of the established aggregation method.

In spite of the limitations mentioned above, the flux integration technique refined in this study is feasible for the estimation of area-averaged fluxes over a heterogeneous land surface. Besides, with abundant flux matrix datasets and high accuracy land cover map, the refined method can achieve the goal of determining the area-averaged ET over an irrigated cropland district.

[revised manuscript text omitted]

---

## Author Comment (AC2) · 19 Jan 2017

Dear Referee #2:

We deeply appreciate for your worthwhile comments and suggestions on our manuscript. According to your comments, and those from Referee #1 and Referee #3, we have carefully revised all sections of the paper (revisions and corrections are marked in red). The point-by-point response to your valuable comments and suggestions are as follows:

Major comments:

1. While the authors' works are valuable, the size of paper is too large, with containing less-important information. I recommend authors to drop entire section 4.4 and the related descriptions available in other sections (section 3.3 etc.)

Response: Thanks for your comments. The major objective of this study is to refine an aggregation method for area-averaged fluxes based on our comprehensive dataset of the HiWATER. The results are also useful for the water balance study extended to the whole Heihe River basin. Thus, the results of area-averaged ET over the study area are still kept in Section 4.4. But some irrelevant parts have been deleted, according to the comments from you and other referees.

2. I could not find a purpose of Fig 8, and comparison with "remotely sensed ET data" (Table 5 and 6).

Response: The purpose of Fig. 8 is to show the spatial pattern of daily ET over the study area for readers. According to comments from yours and other referees, all the relevant statements about the comparison with "remotely sensed ET data" (including Table 5 and 6) and related descriptions in other sections have been deleted.

3. Also, including the comparison with remotely-sensed ET data in this paper might derive another problem on reviewing process because the procedure adopted in the paper is not well described in the paper, and the applied method may not be appropriate.

Response: Thanks. According to your comments and similar comments from other referees, we have dropped all of the relevant parts on the comparison with remotely-sensed ET data in the revised paper.

Minor comments:

Page 8 Line 10: Authors manually revised land cover map using high-resolution CCD images and Google Earth imagery. Do those images applicable for year 2012?

Response: Yes. The CCD images were acquired on 26 July 2012, while the Google

Earth image used was collected on 3 September 2012. Both are in the HiWATER intensive observation period. We have added the acquisition dates of CDD images and the Google earth image in the revised manuscript.

By reviewing the results, the EC data used in the paper seems to be reliable. However, it is better to describe in the paper some more about the measurement accuracy of their EC data, for example, about the energy closure error.

Response: Thank you very much for your suggestion. We have added the descriptions on the data quality of EC and LAS used in HiWATER, as well as the energy balance closure rate, in the revised paper (section 2.2.1).

It is author's preference and authors do not need to change, but I might recommend changing Fig 2-b (ET) from bar-graph with mm/d, to line graph with W/m2 like Fig 2-a, so readers can understand the energy balance condition of the sites by directly comparing Fig 2-a and 2-b.

Response: Accepted.

It's author's preference and authors do not need to change, but Fig.4 can be expressed not as figure but as table.

Response: Thanks for your kind suggestion. However, here, a figure is probably more obvious than a table to show the spatial representativeness of all EC sites. So the original figure is still kept.

In Fig. 6, I recommend authors to show the character "a" "b" "c" and "d" in the figure, because authors are referring the figure such as "fig. 6c" in the text.

Response: Accepted.

Thank you again for your valuable comments and suggestions on our manuscript. The revised manuscript is attached as supplement.

Sincerely yours,
Feinan Xu

Email: xufeinan@lzb.ac.cn

PS. After revising our manuscript and finishing the above responses to you, yesterday, we received the comments from Prof. Thomas Foken (as Referee #4). Some important revisions would be needed based on his comments. A new version might be uploaded within two weeks..

Corresponding author: Weizhen Wang, weizhen@lzb.ac.cn

Northwest Institute of Eco-Environment and Resources, Chinese Academy of Sciences, 320 Donggang west road, Lanzhou, Gansu, 730000, China.

Please also note the supplement to this comment:
http://www.hydrol-earth-syst-sci-discuss.net/hess-2016-602/hess-2016-602-AC2-supplement.pdf
* * *
[Figure]

**Supplement:**

[revised manuscript text omitted]

For the ECs used in our data analysis, the systematic errors had been tried to reduce to a minimum with a pre-observation intercomparison (Xu et al., 2013) and careful maintenance during the observation. The random errors were also analyzed by a separate research (Wang et al., 2015), which can be minimized in an ensemble average. As for the LASs, the errors from data processing processes (e.g. the Bowen-ratio correction problem) were also tried to minimize as much as possible through intercomparisons with fluxes from EC.

For the ECs, the energy balance closure ratio (EBR) was also assessed (Xu et al., 2017). The ratio

of the turbulent heat flux (the sensible and latent heat flux) to available energy for 16 EC sites over vegetated surfaces was 87 %, and the EBR for site 4 was 0.84.

**2.2.2 Collection and processing of remote-sensing data products**

A land cover classification map with 1-m spatial resolution was collected. The map was derived from 
[revised manuscript text omitted]
 values of these input parameters can be derived from measurements generally available from EC flux towers. The daily flux contribution area of the EC flux measurements was calculated by Eq. (8), which provides approximately 90 % of the total source area that contributes to the measured fluxes. Every 30-min flux source area of the LAS sites was estimated via Eq. (7), and the 90 % half-hourly footprint contours of LAS measurements were used. The normalized daily footprint climatology of ECs and half-hourly footprint estimates of LASs were individually overlaid

with 1-m land cover map to determine the footprint-weighted contribution of each land cover classes to the measured flux from EC and LAS systems.

**3.3 Data processing flow of the determination of area-averaged fluxes**

The overall data processing flow for determining the area-averaged evapotranspiration over a heterogeneous land surface mainly includes three aspects (Fig. 2).

Firstly, the spatial representativeness of the 16 EC flux towers within the $5 \times 5$ km$^2$ experiment area was quantitatively assessed by overlaying in-site flux footprint climatology with 1-m land cover map. Detailed analyses on this aspect are going to be presented in the following section.

The second aspect was to evaluate the reliability of the established flux aggregation schemes. The land cover specific flux was firstly dis-aggregated from multiple EC flux measurements by performing a multiple linear regression analysis (Eq. 5). The EC dis-aggregated fluxes of each land cover classes were then aggregated again to obtain area-averaged fluxes, according to the fractional weight of each land cover class in the LAS footprint (Eq. 4). Finally, the aggregated fluxes were compared with LAS observations.

At last, the area-averaged evapotranspiration over a heterogeneous land surface was estimated from multi-point EC flux measurements with the developed flux integration schemes, based on footprint analysis and high resolution land cover map.

**4 Results and Discussion**

**4.1 The characteristics of the surface heat and water vapor fluxes**

Figure 3 depicts the diurnal cycle of the sensible (Fig. 3a) and latent (Fig. 3b) heat fluxes at different sites on two clear days. Both of the two figures reveal significant differences in the magnitude of the sensible- and latent heat fluxes between different surface types during the growing season.

The sensible heat flux over residential area reached a maximum of about 150 W m$^{-2}$ at afternoon

and was higher than that over the vegetated surfaces (H_ec4, Fig. 3a), while the latent heat flux was smaller compared with other sites, with maximum value of less than 300 W m$^{-2}$ due to a certain fraction of sealed land surfaces (LE_ec4, Fig. 3b).

Over the vegetated surfaces (orchard, vegetable, maize), the sensible heat flux was nearly less than 100 W m$^{-2}$ because of well-irrigated cropland (Fig. 3a). The sensible heat flux over the three types of vegetation was also significantly different, and there was also a difference in sensible heat fluxes among maize sites. The mean value of the standard deviation (SD) of H for 13 maize sites was about 8.4 W m$^{-2}$ (Fig. 4a).

Deviations in latent heat fluxes over different vegetation types were also found (Fig. 3b, Fig. 4b). The maize fields performed highly latent heat fluxes and lower sensible heat fluxes than the other two vegetated surfaces. One of the possible reasons is that both of the orchard area and the vegetable field are relatively sparse compared with the maize cropland.

The mean value of the SD of LE for all maize sites was approximately 43.3 W m$^{-2}$. The result showed that the latent heat flux over maize cropland exhibited larger SD than the sensible heat flux, and it also indicated the LE differed between sites for same underlying surface (Fig. 4). This can be partly explained by the discrepancy in plant physiology and vegetation growing stage.

The preliminary results indicated that the variability and difference in the surface energy fluxes between the HiWATER tower flux sites was significant during the crop growth period. The differences in sensible and latent heat fluxes between maize field sites also could be noticed.

**4.2 Analysis of the representativeness of the multi-point EC flux measurements**

To further understand the variability of surface energy fluxes between different sites in a heterogeneous landscape, the footprint analyses for representativeness of EC sites were applied by overlaying flux footprint with high resolution land-cover map (Fig. 1). The fraction of land cover classes present in the daily-averaged footprint of each EC measurements is given in Fig. 5. Given the

source area (90 % flux contribution) of the 4 ECs (sites 5, 8, 13 and 16) on 30 June 2012 exceeded the extent of land cover map, the spatial representative of the 4 EC sites were not shown in Fig. 5b.

Due to the variations in the observation height, atmospheric stability, wind direction and wind speed, the exact shape and size of the EC source area were distinctly different (Fig. 1). For each EC flux measurements, there was more than one type of land cover in its footprint. The contribution of each land cover classes to the total measured flux for EC sites was changed with the varying source area (Fig. 5).

The dominated surface types in the source area were vegetable and orchard at sites 1 and 17, respectively. For site 4, however, there were mainly three types of land cover within its footprint, namely non-vegetation, maize and woods type. The fractional weight of the non-vegetation type and maize field in the footprint greatly varied, while the proportion of woods was almost changeless.

At maize field sites, the relative contribution of maize field to the EC measured flux was approximately more than 0.9, except for sites 2, 9 and 10. At site 2, the percentage of non-vegetation type in the footprint was almost 0.18. For site 9, the rate of maize and non-vegetation type present in footprint significantly varied. The contribution of vegetable type to the flux measurements at site 10 ranged from 0.15 to 0.1.

The above analysis shows that the tower flux measurements at the field scale are generally representative of multiple surface types. The result indicates that the latent and sensible heat fluxes measured by EC systems are representative of the averaged fluxes, which are weighted the upwind surface flux emanating from individual land cover classes with flux footprint. In general, it may be problematic to validate the model estimated fluxes by direct comparison with tower-based flux measurements over a heterogeneous land surface. Thus, the extension of multiple flux measurements to pixel/grid scale surface fluxes is urgently needed.

**4.3 Evaluation of the EC aggregated fluxes**

The determination of area-averaged fluxes from point measurements is usually not straightforward,

especially for heterogeneous land surfaces. Based on multi-point EC flux measurements and accurate 1-m land cover map, a flux aggregation method was established to estimate averaged surface fluxes with footprint analysis and multivariate regression. Fig. 1 shows that all types of land covers present in the LAS flux footprint, so the LAS measurements can be taken as reference to assess the feasibility of the developed integration schemes.

At first, the sensible heat flux for each land cover was dis-aggregated from the EC observed component fluxes in a heterogeneous footprint with multiple linear regression method. The diurnal cycle of the EC dis-aggregated sensible heat fluxes for each land cover types is highly significant (Fig. 6). During the crop growth stage, the sensible heat flux over maize field showed a minimum value in the afternoon, while the sensible heat fluxes for non-vegetation type at daytime exhibited a maximum of about 200 W m$^{-2}$. The sensible heat fluxes for vegetable and woods types lied between them.

Then, the sensible heat flux representative for the LAS source area was aggregated by multiplying the EC dis-aggregated fluxes for the four land-cover classes by their relative fraction in the LAS source area. Fig. 7 illustrates a scatterplot of 30-min averaged sensible heat fluxes estimated using the flux aggregation method (hereafter referred as H_ECagg) versus LAS measurements (H_LAS), as well as the linear regression parameters (including equations and R$^2$). The different statistics between LAS observed fluxes and EC aggregated results are listed in Table 3.

For LAS 1 (see Fig. 7a and Table 3), a good agreement is found between EC aggregated fluxes and LAS measurements, with high correlation coefficient and low RMSE value (R$^2$= 0.79, RMSE= 0.96 W m$^{-2}$). The scatter points in the graph are nearly close to the 1:1 line. The MBE and MAPE values were 4.25 W m$^{-2}$ and 9.93 %, respectively.

Compared with LAS 1, there was a little scatter between LAS measured fluxes and estimates from multiple EC flux observations for LAS 2, but yielding a small mean bias error (MBE = 2.31 W m$^{-2}$) (Fig. 7b, Table 3). RMSE and MAPE values between H_ECagg and H_LAS2 were much higher than that of LAS 1, with values of 6.91 W m$^{-2}$ and 16.39 %, respectively. Considering the heterogeneous

distribution of surface covers in the LAS source area, slight area of non-vegetation distributing in the center of LAS 2 path would be the primary factor attributing to the bias (blue circles in Fig. 1).

For LAS 3 (Fig. 7c, Table 3), there was a slightly weak relationship between sensible heat fluxes derived from the LAS measurements and flux aggregation method, with correlation coefficient ($R^2$) of 0.57 and RMSE, MAPE and MBE values of 17.63 W m$^{-2}$, 31.7 % and -18.01 W m$^{-2}$, respectively. The scatter points in Fig. 7c were overall below the 1:1 line. It indicated that the 30-min averaged H estimated from EC flux observations using the aggregation method were underestimated against LAS derived H (negative MBE). As shown in Fig. 1, there is more large area of residential areas randomly distributing in the center of LAS 3 path than other three LAS systems. This discrepancy is likely related to the heterogeneously distributed surface types.

In Fig. 7d, the area-averaged sensible heat fluxes obtained using the flux aggregation method were consistent with LAS measurements, with $R^2$ of 0.57 for LAS 4. In contrast with LAS 3, the scatter points in this graph were almost above the 1:1 line (overestimate of EC estimated H, MBE > 10 W m$^{-2}$). RMSE value of LAS 4 relatively decreased by 4.88 W m$^{-2}$, but MAPE value was up to 33.7 %. The red open squares in Fig. 7d are more close to the 1:1 line than the blue open circles. When southeast wind prevailed, the relative contribution of non-vegetation type to the LAS 4 measurements was about 0.2, and its value decreased to 0.08 when the main wind direction was northwest.

Through detailed analysis, the magnitude of divergences between the estimated and measured area-averaged surface fluxes is in large part concerned with the variation of corresponding LAS source area. Moreover, the contribution of non-vegetation type to the LAS observations, which accounted for a large proportion in the footprint of LASs (LAS 2, 3 and 4), would be one of the main factors contributing to the bias between the estimated results and LAS measurements. The reason is that the EC dis-aggregated sensible heat flux for non-vegetation type may not be representative for the flux emanating from sealed buildings and roads that are part of non-vegetation type. On the other hand, the complexity of the LAS footprints may lead to the inconsistency of EC-estimated and LAS-measured

averaged fluxes.

Overall, the above results demonstrate that, compared with the area-averaged fluxes measured by LAS systems, the area-averaged fluxes that are aggregated from multiple EC flux measurements using the established flux aggregation method are reliable. Therefore, the developed flux integration schemes in this study can be an effective way to estimate areal averaged fluxes.

**4.4 Estimation of area-averaged evapotranspiration**

The flux aggregation scheme, which was established and evaluated in Sect. 4.3, was adopted to determine the area-averaged ET over our study area with multi-point EC flux measurements and high resolution land-cover map. The EC dis-aggregated daily ET for all the land covers over two clear days was shown in Fig. 8. As can be seen, the daily ET values for maize field were highest during the crop growing season (7 mm – 8 mm). The value of daily ET was 6.4 mm for woods type, and it ranged from 6 mm to 7 mm for vegetable type. On the contrary, the daily ET for non-vegetation type varied largely, with values of 2.8 mm on 29 June and 1.5 mm on 30 June, respectively.

The daily ET maps at 1-m resolution were produced through the dis-aggregated daily ET for all land cover classes, combined with the 1-m land classification map. Fig. 9 depicts the spatial pattern of daily ET on 29 and 30 June 2012. It can be seen from the legend in figure, the daily ET ranged from 1.56 to 7.95 mm during the two days, and with higher values on 29 June (Fig. 9a) for all land cover classes than that on 30 June (Fig. 9b). The maize field performed highest ET value and distributed widely, whereas other three types of land cover randomly distributed across the whole study area with quite different ET values.

Table 4 lists the total ET for different land cover classes and their proportion of the total area ET. The total ET for our study area was almost 169626.77 m$^3$ on 29 June, while it was about 152948.41 m$^3$ on 30 June. The results demonstrated that the ratio of ET for maize field to the total area ET was in excess of 80 %. In addition, the total rate of ET for both woods and vegetables types was approximately

13 %, and the ET value for non-vegetation type accounted for 4.83 % of daily totals on the average.

Finally, the area-averaged daily ET over the kernel experiment area of HiWATER was estimated via Eq. (1), with values of approximately 7.01 mm on 29 June and 6.32 mm on 30 June 2012.

**5 Summary and conclusions**

On the basis of 1-m accurate land cover map and multi-point ground-based flux measurements datasets from 16 EC systems and 4 groups of LAS systems during the intensive observation period of HiWATER program, the area-averaged surface fluxes over a heterogeneous surface were determined by a flux aggregation method, which was established through the integration of footprint analysis and multiple regression. The estimated area-averaged fluxes were validated by the LAS measurements to assess the reliability of the integration method. Ultimately, the method that had been evaluated was used to estimate area-averaged ET over our study area.

First and foremost, analyses of the spatial representativeness of multiple EC flux towers were performed for the interpretation of the surface fluxes over different land surfaces. It is proved that the combination of footprint analysis and high-resolution land cover map can be a proper way to clarify the relationship between the tower-based flux observations over heterogeneous surfaces and individual land cover specific fluxes, and it is also the foundation for the establishment of flux aggregation scheme.

Secondly, based on good multi-scale (EC & LAS) flux datasets, precise flux footprints of flux towers and better land cover classification map, a flux aggregation scheme can be successfully established through the integration of footprint analysis and multivariate regression. In a heterogeneous study area, the surface flux emanating from individual land cover classes need to be firstly acquired by a flux integration method before deriving area-averaged fluxes. The results show that the developed flux aggregation method provides a unique opportunity to disentangle the heterogeneous land surface fluxes in their single components.

Then, the averaged surface fluxes estimated from the established flux aggregation method were

compared with the corresponding observed LAS values. The MAPE values of the LAS 1, LAS 2, LAS 3 and LAS 4 were 9.93 %, 16.39 %, 31.7 % and 33.7 %, respectively. The reasons for the divergences between EC aggregated estimates and LAS observations were investigated by a combination of remote sensing data and ground measurements, and the findings revealed that the extent of vegetation structure heterogeneity had a significant influence on the application of the established aggregation method.

In spite of the limitations mentioned above, the flux integration technique refined in this study is feasible for the estimation of area-averaged fluxes over a heterogeneous land surface. Besides, with abundant flux matrix datasets and high accuracy land cover map, the refined method can achieve the goal of determining the area-averaged ET over an irrigated cropland district.

[revised manuscript text omitted]

---

## Author Comment (AC3) · 19 Jan 2017

Dear Referee # 3:

We appreciate very much for your valuable comments and suggestions on our manuscript. According to your comments and those from Referee #1 and Referee #2, we have carefully revised all sections of the paper (revisions and corrections are marked in red). Detailed response to your worthwhile comments and suggestions are as follows:

General comments:

1. To address the scientific problem in this paper, 30-min flux might be sufficient, given the uncertainty in gap-filling method (rainfall, fog etc). Daily ET (section 4.4) does not help a lot here. Indeed, it might be more necessary to clarify data quality and uncertainty of the EC and LAS measurements.

Response: Thanks for your comments. Yes, by giving the uncertainty in gap-filling, the 30-min flux might be sufficient to address the scientific problem in this paper. However, the major objective of this study, besides refining the aggregation method for area averaged fluxes (based on our unique and comprehensive dataset of the HiWATER), is also finally useful for the water balance study extended to the whole Heihe River basin. So the area-averaged daily ET over the study area is added for reference. We have also clarified the data quality and uncertainties of the EC and LAS measurements in Section 2.1.1 of the revised paper.

2. Besides, P-M estimated ET could be removed.

Response: Yes. According to your comments (and those from other referees), we have removed the descriptions relevant to P-M estimated ET, such as those in Section 2.2.2, Section 3.3, and the paragraph on the comparison with P-M ET in Section 4.4.

Specific comments:

Page 5 Line 24 "following Fig.3" Since it is the first figure appearing in this article, it's better to change the number from 3 to 1.

Response: Accepted.

Page 6 Line 15 "EC data from 16 towers. . ." According to section 4.2, in addition to site 3, sites 5/8/13/16 were also not used. It is better to use a consistent dataset throughout the paper.

Response: We are sorry for the unclear statements about the data used in the paper. EC data from 16 towers were all used for analysis. We have improved our statements in Section 4.2 of the revised manuscript.

Page 6 Line 16 "no irrigation" and how was the weather during the period?

Response: To choose "no irrigation" days is mainly for reducing the effect of local advection. The two days, 29 and 30 June 2012, were typical clear days.

Page 6 Line 22 "coordinate rotation" why not use Planar Fit?

Response: Our study area, the oasis in the middle reaches of Heihe River basin, is relatively very flat. To use the common 2-D rotation method is not only simpler but also enough in this situation. Actually, we have compared the results of 2-D rotation and Planar Fit during previous works. The differences were very small.

Page 7 Line 13 "MOST" there are different solutions. Add either corresponding equations or references here.

Response: A reference had been added: Andreas, E. L.: Estimating Cn2 over snow and sea ice from meteorological data, JOSA A, 5, 481-495, 1988.

And how were roughness height and zero-plane displacement estimated?

Response: The roughness height and zero-plane displacement of the study area were obtained following Martano (2000). This has been added in the revised manuscript. Martano, P.: Estimation of Surface Roughness Length and Displacement Height from Single-Level Sonic Anemometer Data, Journal of Applied Meteorology, 39, 708-715, 2000.

Page 7 Line 16 'daytime...' It's a bit confusing. Local time is better.

Response: We have stated the time difference between Beijing Standard Time and Local time in the revised paper.

For data quality control, what is the threshold value of signal strength?

Response: For BLS series (BLS4500/BLS900), the threshold value of signal strength is 1000.

Section 2.2.2 This section could be abbreviated if the preliminary land cover has already been done by Liu and Bo (2015).

Response: Accepted.

Page 8 Line 11, specify the date of the google earth image.

Response: The Google Earth image used was collected on 3 September 2012. This has been added in the revision.

Page 8 Line 16-20, it might not be necessary to compare with PM-ET. The principle of that is the same as the comparison with LAS in terms of flux aggregation and there might uncertainty in PM-ET.

Response: As mentioned in the beginning of this reply, we have accepted this comment and made revision.

Section 4.3 Page 15 &16. It's better to look into the details to figure out the factors contributing to the bias between EC and LAS, instead of just mentioning 'heterogeneous distribution of surface covers'.

Response: Accepted.

Section 4.4 I didn't see the difference between Table 5 and 6 in terms of addressing the problem despite their different units.

Response: We have removed all the related text on the comparison with PM-ET (including Table 6 and Table 5) in section 4.4 of the revised manuscript.

Thank you again for your valuable comments and suggestions on our manuscript. The revised manuscript is attached as supplement.

Sincerely yours,

Feinan Xu

Email: xufeinan@lzb.ac.cn

PS. After revising our manuscript and finishing the above responses to you, yesterday, we received the comments from Prof. Thomas Foken (as Referee #4). Some important revisions would be needed based on his comments. A new version might be uploaded within two weeks.

Corresponding author: Weizhen Wang, weizhen@lzb.ac.cn

Northwest Institute of Eco-Environment and Resources, Chinese Academy of Sciences, 320 Donggang west road, Lanzhou, Gansu, 730000, China.

Please also note the supplement to this comment:
http://www.hydrol-earth-syst-sci-discuss.net/hess-2016-602/hess-2016-602-AC3-supplement.pdf

**Supplement:**

[revised manuscript text omitted]

For the ECs used in our data analysis, the systematic errors had been tried to reduce to a minimum with a pre-observation intercomparison (Xu et al., 2013) and careful maintenance during the observation. The random errors were also analyzed by a separate research (Wang et al., 2015), which can be minimized in an ensemble average. As for the LASs, the errors from data processing processes (e.g. the Bowen-ratio correction problem) were also tried to minimize as much as possible through intercomparisons with fluxes from EC.

For the ECs, the energy balance closure ratio (EBR) was also assessed (Xu et al., 2017). The ratio

of the turbulent heat flux (the sensible and latent heat flux) to available energy for 16 EC sites over vegetated surfaces was 87 %, and the EBR for site 4 was 0.84.

**2.2.2 Collection and processing of remote-sensing data products**

A land cover classification map with 1-m spatial resolution was collected. The map was derived from 
[revised manuscript text omitted]
 values of these input parameters can be derived from measurements generally available from EC flux towers. The daily flux contribution area of the EC flux measurements was calculated by Eq. (8), which provides approximately 90 % of the total source area that contributes to the measured fluxes. Every 30-min flux source area of the LAS sites was estimated via Eq. (7), and the 90 % half-hourly footprint contours of LAS measurements were used. The normalized daily footprint climatology of ECs and half-hourly footprint estimates of LASs were individually overlaid

with 1-m land cover map to determine the footprint-weighted contribution of each land cover classes to the measured flux from EC and LAS systems.

**3.3 Data processing flow of the determination of area-averaged fluxes**

The overall data processing flow for determining the area-averaged evapotranspiration over a heterogeneous land surface mainly includes three aspects (Fig. 2).

Firstly, the spatial representativeness of the 16 EC flux towers within the $5 \times 5$ km$^2$ experiment area was quantitatively assessed by overlaying in-site flux footprint climatology with 1-m land cover map. Detailed analyses on this aspect are going to be presented in the following section.

The second aspect was to evaluate the reliability of the established flux aggregation schemes. The land cover specific flux was firstly dis-aggregated from multiple EC flux measurements by performing a multiple linear regression analysis (Eq. 5). The EC dis-aggregated fluxes of each land cover classes were then aggregated again to obtain area-averaged fluxes, according to the fractional weight of each land cover class in the LAS footprint (Eq. 4). Finally, the aggregated fluxes were compared with LAS observations.

At last, the area-averaged evapotranspiration over a heterogeneous land surface was estimated from multi-point EC flux measurements with the developed flux integration schemes, based on footprint analysis and high resolution land cover map.

**4 Results and Discussion**

**4.1 The characteristics of the surface heat and water vapor fluxes**

Figure 3 depicts the diurnal cycle of the sensible (Fig. 3a) and latent (Fig. 3b) heat fluxes at different sites on two clear days. Both of the two figures reveal significant differences in the magnitude of the sensible- and latent heat fluxes between different surface types during the growing season.

The sensible heat flux over residential area reached a maximum of about 150 W m$^{-2}$ at afternoon

and was higher than that over the vegetated surfaces (H_ec4, Fig. 3a), while the latent heat flux was smaller compared with other sites, with maximum value of less than 300 W m$^{-2}$ due to a certain fraction of sealed land surfaces (LE_ec4, Fig. 3b).

Over the vegetated surfaces (orchard, vegetable, maize), the sensible heat flux was nearly less than 100 W m$^{-2}$ because of well-irrigated cropland (Fig. 3a). The sensible heat flux over the three types of vegetation was also significantly different, and there was also a difference in sensible heat fluxes among maize sites. The mean value of the standard deviation (SD) of H for 13 maize sites was about 8.4 W m$^{-2}$ (Fig. 4a).

Deviations in latent heat fluxes over different vegetation types were also found (Fig. 3b, Fig. 4b). The maize fields performed highly latent heat fluxes and lower sensible heat fluxes than the other two vegetated surfaces. One of the possible reasons is that both of the orchard area and the vegetable field are relatively sparse compared with the maize cropland.

The mean value of the SD of LE for all maize sites was approximately 43.3 W m$^{-2}$. The result showed that the latent heat flux over maize cropland exhibited larger SD than the sensible heat flux, and it also indicated the LE differed between sites for same underlying surface (Fig. 4). This can be partly explained by the discrepancy in plant physiology and vegetation growing stage.

The preliminary results indicated that the variability and difference in the surface energy fluxes between the HiWATER tower flux sites was significant during the crop growth period. The differences in sensible and latent heat fluxes between maize field sites also could be noticed.

**4.2 Analysis of the representativeness of the multi-point EC flux measurements**

To further understand the variability of surface energy fluxes between different sites in a heterogeneous landscape, the footprint analyses for representativeness of EC sites were applied by overlaying flux footprint with high resolution land-cover map (Fig. 1). The fraction of land cover classes present in the daily-averaged footprint of each EC measurements is given in Fig. 5. Given the

source area (90 % flux contribution) of the 4 ECs (sites 5, 8, 13 and 16) on 30 June 2012 exceeded the extent of land cover map, the spatial representative of the 4 EC sites were not shown in Fig. 5b.

Due to the variations in the observation height, atmospheric stability, wind direction and wind speed, the exact shape and size of the EC source area were distinctly different (Fig. 1). For each EC flux measurements, there was more than one type of land cover in its footprint. The contribution of each land cover classes to the total measured flux for EC sites was changed with the varying source area (Fig. 5).

The dominated surface types in the source area were vegetable and orchard at sites 1 and 17, respectively. For site 4, however, there were mainly three types of land cover within its footprint, namely non-vegetation, maize and woods type. The fractional weight of the non-vegetation type and maize field in the footprint greatly varied, while the proportion of woods was almost changeless.

At maize field sites, the relative contribution of maize field to the EC measured flux was approximately more than 0.9, except for sites 2, 9 and 10. At site 2, the percentage of non-vegetation type in the footprint was almost 0.18. For site 9, the rate of maize and non-vegetation type present in footprint significantly varied. The contribution of vegetable type to the flux measurements at site 10 ranged from 0.15 to 0.1.

The above analysis shows that the tower flux measurements at the field scale are generally representative of multiple surface types. The result indicates that the latent and sensible heat fluxes measured by EC systems are representative of the averaged fluxes, which are weighted the upwind surface flux emanating from individual land cover classes with flux footprint. In general, it may be problematic to validate the model estimated fluxes by direct comparison with tower-based flux measurements over a heterogeneous land surface. Thus, the extension of multiple flux measurements to pixel/grid scale surface fluxes is urgently needed.

**4.3 Evaluation of the EC aggregated fluxes**

The determination of area-averaged fluxes from point measurements is usually not straightforward,

especially for heterogeneous land surfaces. Based on multi-point EC flux measurements and accurate 1-m land cover map, a flux aggregation method was established to estimate averaged surface fluxes with footprint analysis and multivariate regression. Fig. 1 shows that all types of land covers present in the LAS flux footprint, so the LAS measurements can be taken as reference to assess the feasibility of the developed integration schemes.

At first, the sensible heat flux for each land cover was dis-aggregated from the EC observed component fluxes in a heterogeneous footprint with multiple linear regression method. The diurnal cycle of the EC dis-aggregated sensible heat fluxes for each land cover types is highly significant (Fig. 6). During the crop growth stage, the sensible heat flux over maize field showed a minimum value in the afternoon, while the sensible heat fluxes for non-vegetation type at daytime exhibited a maximum of about 200 W m$^{-2}$. The sensible heat fluxes for vegetable and woods types lied between them.

Then, the sensible heat flux representative for the LAS source area was aggregated by multiplying the EC dis-aggregated fluxes for the four land-cover classes by their relative fraction in the LAS source area. Fig. 7 illustrates a scatterplot of 30-min averaged sensible heat fluxes estimated using the flux aggregation method (hereafter referred as H_ECagg) versus LAS measurements (H_LAS), as well as the linear regression parameters (including equations and R$^2$). The different statistics between LAS observed fluxes and EC aggregated results are listed in Table 3.

For LAS 1 (see Fig. 7a and Table 3), a good agreement is found between EC aggregated fluxes and LAS measurements, with high correlation coefficient and low RMSE value (R$^2$= 0.79, RMSE= 0.96 W m$^{-2}$). The scatter points in the graph are nearly close to the 1:1 line. The MBE and MAPE values were 4.25 W m$^{-2}$ and 9.93 %, respectively.

Compared with LAS 1, there was a little scatter between LAS measured fluxes and estimates from multiple EC flux observations for LAS 2, but yielding a small mean bias error (MBE = 2.31 W m$^{-2}$) (Fig. 7b, Table 3). RMSE and MAPE values between H_ECagg and H_LAS2 were much higher than that of LAS 1, with values of 6.91 W m$^{-2}$ and 16.39 %, respectively. Considering the heterogeneous

distribution of surface covers in the LAS source area, slight area of non-vegetation distributing in the center of LAS 2 path would be the primary factor attributing to the bias (blue circles in Fig. 1).

For LAS 3 (Fig. 7c, Table 3), there was a slightly weak relationship between sensible heat fluxes derived from the LAS measurements and flux aggregation method, with correlation coefficient ($R^2$) of 0.57 and RMSE, MAPE and MBE values of 17.63 W m$^{-2}$, 31.7 % and -18.01 W m$^{-2}$, respectively. The scatter points in Fig. 7c were overall below the 1:1 line. It indicated that the 30-min averaged H estimated from EC flux observations using the aggregation method were underestimated against LAS derived H (negative MBE). As shown in Fig. 1, there is more large area of residential areas randomly distributing in the center of LAS 3 path than other three LAS systems. This discrepancy is likely related to the heterogeneously distributed surface types.

In Fig. 7d, the area-averaged sensible heat fluxes obtained using the flux aggregation method were consistent with LAS measurements, with $R^2$ of 0.57 for LAS 4. In contrast with LAS 3, the scatter points in this graph were almost above the 1:1 line (overestimate of EC estimated H, MBE > 10 W m$^{-2}$). RMSE value of LAS 4 relatively decreased by 4.88 W m$^{-2}$, but MAPE value was up to 33.7 %. The red open squares in Fig. 7d are more close to the 1:1 line than the blue open circles. When southeast wind prevailed, the relative contribution of non-vegetation type to the LAS 4 measurements was about 0.2, and its value decreased to 0.08 when the main wind direction was northwest.

Through detailed analysis, the magnitude of divergences between the estimated and measured area-averaged surface fluxes is in large part concerned with the variation of corresponding LAS source area. Moreover, the contribution of non-vegetation type to the LAS observations, which accounted for a large proportion in the footprint of LASs (LAS 2, 3 and 4), would be one of the main factors contributing to the bias between the estimated results and LAS measurements. The reason is that the EC dis-aggregated sensible heat flux for non-vegetation type may not be representative for the flux emanating from sealed buildings and roads that are part of non-vegetation type. On the other hand, the complexity of the LAS footprints may lead to the inconsistency of EC-estimated and LAS-measured

averaged fluxes.

Overall, the above results demonstrate that, compared with the area-averaged fluxes measured by LAS systems, the area-averaged fluxes that are aggregated from multiple EC flux measurements using the established flux aggregation method are reliable. Therefore, the developed flux integration schemes in this study can be an effective way to estimate areal averaged fluxes.

**4.4 Estimation of area-averaged evapotranspiration**

The flux aggregation scheme, which was established and evaluated in Sect. 4.3, was adopted to determine the area-averaged ET over our study area with multi-point EC flux measurements and high resolution land-cover map. The EC dis-aggregated daily ET for all the land covers over two clear days was shown in Fig. 8. As can be seen, the daily ET values for maize field were highest during the crop growing season (7 mm – 8 mm). The value of daily ET was 6.4 mm for woods type, and it ranged from 6 mm to 7 mm for vegetable type. On the contrary, the daily ET for non-vegetation type varied largely, with values of 2.8 mm on 29 June and 1.5 mm on 30 June, respectively.

The daily ET maps at 1-m resolution were produced through the dis-aggregated daily ET for all land cover classes, combined with the 1-m land classification map. Fig. 9 depicts the spatial pattern of daily ET on 29 and 30 June 2012. It can be seen from the legend in figure, the daily ET ranged from 1.56 to 7.95 mm during the two days, and with higher values on 29 June (Fig. 9a) for all land cover classes than that on 30 June (Fig. 9b). The maize field performed highest ET value and distributed widely, whereas other three types of land cover randomly distributed across the whole study area with quite different ET values.

Table 4 lists the total ET for different land cover classes and their proportion of the total area ET. The total ET for our study area was almost 169626.77 m$^3$ on 29 June, while it was about 152948.41 m$^3$ on 30 June. The results demonstrated that the ratio of ET for maize field to the total area ET was in excess of 80 %. In addition, the total rate of ET for both woods and vegetables types was approximately

13 %, and the ET value for non-vegetation type accounted for 4.83 % of daily totals on the average.

Finally, the area-averaged daily ET over the kernel experiment area of HiWATER was estimated via Eq. (1), with values of approximately 7.01 mm on 29 June and 6.32 mm on 30 June 2012.

**5 Summary and conclusions**

On the basis of 1-m accurate land cover map and multi-point ground-based flux measurements datasets from 16 EC systems and 4 groups of LAS systems during the intensive observation period of HiWATER program, the area-averaged surface fluxes over a heterogeneous surface were determined by a flux aggregation method, which was established through the integration of footprint analysis and multiple regression. The estimated area-averaged fluxes were validated by the LAS measurements to assess the reliability of the integration method. Ultimately, the method that had been evaluated was used to estimate area-averaged ET over our study area.

First and foremost, analyses of the spatial representativeness of multiple EC flux towers were performed for the interpretation of the surface fluxes over different land surfaces. It is proved that the combination of footprint analysis and high-resolution land cover map can be a proper way to clarify the relationship between the tower-based flux observations over heterogeneous surfaces and individual land cover specific fluxes, and it is also the foundation for the establishment of flux aggregation scheme.

Secondly, based on good multi-scale (EC & LAS) flux datasets, precise flux footprints of flux towers and better land cover classification map, a flux aggregation scheme can be successfully established through the integration of footprint analysis and multivariate regression. In a heterogeneous study area, the surface flux emanating from individual land cover classes need to be firstly acquired by a flux integration method before deriving area-averaged fluxes. The results show that the developed flux aggregation method provides a unique opportunity to disentangle the heterogeneous land surface fluxes in their single components.

Then, the averaged surface fluxes estimated from the established flux aggregation method were

compared with the corresponding observed LAS values. The MAPE values of the LAS 1, LAS 2, LAS 3 and LAS 4 were 9.93 %, 16.39 %, 31.7 % and 33.7 %, respectively. The reasons for the divergences between EC aggregated estimates and LAS observations were investigated by a combination of remote sensing data and ground measurements, and the findings revealed that the extent of vegetation structure heterogeneity had a significant influence on the application of the established aggregation method.

In spite of the limitations mentioned above, the flux integration technique refined in this study is feasible for the estimation of area-averaged fluxes over a heterogeneous land surface. Besides, with abundant flux matrix datasets and high accuracy land cover map, the refined method can achieve the goal of determining the area-averaged ET over an irrigated cropland district.

[revised manuscript text omitted]

---

## Author Comment (AC4) · 12 Feb 2017

Dear Prof. Dr. Thomas Foken:

We appreciate very much for your valuable comments and suggestions on our manuscript. According to your comments, and those from other three referees, we have carefully revised all sections of the paper (revisions and corrections are marked in red). Detailed response to your worthwhile comments and suggestions are as follows:

Major comments:

1. Reading the manuscript, I found that the concept of the experimental design and the data analysis is very similar to the experiment LITFASS-2003, which was published in BAMS (Mengelkamp et al., 2006) and in a special issue of Boundary-Layer Meteorology (2006, vol. 121, issue 1). Some of these papers are quoted, but papers published later are missing (Foken et al., 2006; Foken et al., 2010; Charuchittipan et al., 2014).

Response:

Thanks. We have added the important references you specified.

2. Several parts in the paper are unclear, or information is missing that would enable the paper to be followed accurately:

(1) The area of investigation was very much dominated by maize fields. Only three stations had another dominant land cover (stations 1, 4, and 17). This is a significant limitation for the stated aim of the paper to determine area-averaged fluxes over a heterogeneous area. For the LITFASS-2003 experiment (and other experiments given as references), different land cover types were much better distributed. This deficit should be discussed.

Response:

As described in Section 2.1, even the dominant land-use type in the intensive observation area was maize field, the surface status of this oasis were actually very heterogeneous. The small square maize fields were all staggered with windbreak trees, roads, irrigation ditches, etc. We have classified four dominant types of the land-cover in the study area. The proportions of each land cover classes were 72 % (maize), 15% (non-vegetation), 8 % (woods) and 5 % (vegetable), respectively. According to the crop planting structure and land cover, 13 sites were spatially distributed under the dominated maize cropland; while only three stations, namely site 1 (vegetable field) and site 4 (residential area) as well as site 17 (orchard), were separately installed in respectively rather small area of land-use. This has been discussed in more detail in

the revised manuscript.

(2) The function of the LAS in the aggregation schema was not clear. I could not find a reason for the use of such data. In LITFASS-2003, LAS systems were also used with a specific function: It was assumed that LAS can also measure the fluxes of larger turbulence or circulation structures and that this is not affected by the non-closure of the energy balance (Foken, 2008). This information was used to discuss the unclosed energy balance of the flux measurements and to correct this. The problem of the unclosed energy balance is not mentioned in the whole paper, but it is a standard for the analysis of surface flux measurements (Foken et al., 2012).

Response:

The LAS measurements for this paper are an intermediate point in checking the established flux aggregation algorithm. The procedure is as follows: the sensible heat fluxes representative for LAS source area were firstly integrated from multiple EC flux measurements, and then compared with the sensible heat fluxes from the 4 paths of LAS systems, to test the reliability of the developed flux integration method. Finally, the latent heat fluxes (daily evapotranspiration) of EC systems were extended to the study area using the aggregation scheme.

The energy balance closure (EBC) is a significant problem we are concerning from the very beginning of HiWATER. Moreover, relevant research has just published in JAMC (Xu et al., 2017). Generally, the energy balance closure ratio (EBR) during the 3 and half months was good. For the 17 EC stations in the intensive observation area, the average EBR was about 0.92. Except the lowest (0.78) in orchard site (#17), values in other sites were scattered without clear relation to the surface status. Site #15 (Super-station) had 2 heights, 4.5 m and 34 m. The relevant EBR were 0.89 & 1.03 respectively. This is quite reasonable.

We have added the detailed description on the EBR for the EC data of the HiWATER flux matrix in Section 2.2.1 and inserted a referenced here. We have discussed the

effect of the unclosed energy balance in EC flux measurements on the results of the flux aggregation method in the revised manuscript.

(3) Any information is missing as to why the footprint model by Kormann and Meixner (2001) was used in your study. Perhaps the textbook by Leclerc and Foken (2014) would give you the relevant information. Questionable is the exact location of the small non-maize-covered areas in the footprint of the EC and LAS measurements. A discussion of the accuracy of the footprint analysis combined with the accuracy of the EC and LAS measurements is urgently necessary.

Response:

The advantage of the analytical footprint model by Kormann and Meixner (2001) was referenced in the textbook by Leclerc and Foken (2014). Related descriptions have been added into our revised manuscript. Besides, as we have checked, the footprint estimates of the Kormann and Meixner (2001) were in good agreement with the results of sophisticated backward Lagrangian footprint models, such as the Kljun scheme (Kljun et al., 2002;Kljun et al., 2015). The results from the newest version of Kljun's scheme (October 2016) was used to compare with what from that of Kormann and Meixner (2001). The differences were really minor. We have added some statements and relevant references in the revised paper.

The land-cover map used in the study was initially derived from the aircraft remote sensing images with 1-m spatial resolution, and was then carefully post-processed. Thus, overlapping the accurate 1-m land-cover map with the footprint of EC and LAS with same resolution can determine the location of the small non-vegetation areas in the footprint.

Quality-control and uncertainty-estimation for the EC and LAS data of the HiWATER flux matrix were carefully done. For the EC systems used in the data analysis, we have tried to reduce the systematic errors to a minimum with a pre-observation inter-comparison and careful maintenances during the observation period (Xu et al., 2013).

The random errors were also analyzed by a separate research, which can be minimized in an ensemble average (Wang et al., 2015). As for the eddy-covariance systems, flux data from the 4 paths of LAS were also quality controlled. The systematic errors from data processing, e.g. the larger effects of Bowen-ratio correction in this oasis area, were carefully minimized. We checked the sensible heat fluxes (H) from the 4 paths of LAS with that from the nearer ECs. Except LAS 3, under its path there are clearly some village buildings so the H_las is higher, others agreed very well with that of ECs. Relevant statements on this have been added into Section 2.1.1.

(4) The applied multiple-linear regression analysis needs more information. Did you aggregate the fluxes according to the land-cover type in different effect levels of the footprint? Compare your method with the methods presented by Leclerc and Foken (2014).

Response:

Yes, we aggregate the fluxes according to the land-cover type in different effect levels of the footprint. We have supplemented some statements on the applied multiple-linear regression method in the study into Section 3.1.

As mentioned above, we have compared carefully the footprint results from Korman and Meixner (2001) with those from Kljun's scheme (Kljun et al. 2002, 2015) to insure the quality of our footprint analysis.

(5) What is meant by "Remotely sensed ET products"? If I understood the paper correctly, only the land-cover type was determined by satellite measurements, but, as seems probable, did these also include the net radiation for use in the Penman-Monteith equation? But this would then be difficult for the heterogeneous land cover. It is impossible to discuss the underestimation of the fluxes by the Penman-Monteith equation without knowing the parameterizations used in this equation. E.g., the atmospheric resistance and the stomata resistance are extremely variable and should be included in any discussion.

Response:

We benefit a lot from your valuable comments. We have removed the parts on comparison with remotely sensed ET products derived by Penman-Monteith equation, according to the comments from you and other referees.

(6) Please also show in Fig. 2 the daily cycle of the evapotranspiration and not only the daily sum. This is necessary to indicate the energy exchange of the different sites, possible oasis effects, and the Bowen ratio. The latter may be a good indicator which to classify the sites.

Response:

Accepted. We have changed the Fig. 2(b) (Fig. 3(b) in revised paper) from bar-graph with mm/d to line graph with W m-2, and also have re-stated the descriptions on the energy exchange of the different sites.

(7) Undoubtedly the authors have an interesting data set with a significant scientific potential. Such a data set should be published with a good scientific concept. Besides some deficits in the experimental design, the concept of area-averaged fluxes may be such a concept. But the paper needs significant improvements according to the points given above. Therefore I recommend major revisions.

Response:

Thanks for your constructive comments.

Minor remarks:

The numbering of the figures is confusing. Figure 3 should be renamed as Fig. 1.

Response:

Accepted.

Table 1: The instrumentation (sonic anemometer, gas analyzer) is missing.

Response:

Accepted.

Table 2: Do not mix LAS type and LAS producer, please give both for all sites.

Response:

Accepted.

p. 6, line 21: What do the flags mean?

Response:

The flag 0, 1 and 2 represent high-quality, intermediate-quality and poor-quality flux data (Mauder and Foken, 2015), respectively. We have added this reference in the revised manuscript.

p. 6, line 23: Why did you use 2D-rotation and not planar fit? Was the terrain absolutely even?

Response:

Our study area, the oasis in the middle reaches of Heihe River basin, is relatively very flat. To use the common 2-D rotation method is not only simpler but also enough in this situation. We have compared the results of 2-D rotation and Planar fit during previous data-processing works. The differences were very small.

p. 7, line 13: L can be easily misinterpreted as the Obukhov length in a micrometeorological paper.

Response:

We have changed the symbol L to R.

Fig. 4 and 5: Why did you use different names or land cover types in both figures?

Response:
We are sorry for our mistake. We have unified the use of land cover types in both figures.

Fig. 6: Probably y has a lower accuracy than given in the figure!

Response:

The different statistics (e.g. the root mean square error, RMSE) between x and y listed in Table 3 were calculated with data shown in the Fig. 7 of the revised manuscript.

p. 16, line 11: The reference should probably be Fig. 3!

Response:

Accepted.

p. 17, line 12: This is trivial; when maize dominates the land cover it is normal that maize also dominates the ET.

Response:

Removed.

Table 6: Give the units in the columns.

Response:

The relevant information on the comparison with P-M estimated ET throughout the paper has already been removed, including Table 6.

p. 19, line 16-25: Such a paper needs a well-written conclusion chapter and not only ten not very significant lines.

Response:

The conclusions have been re-written.

p. 22, line 13: Many authors are missing
Response:

Accepted.

p. 22, line 18: Print CO2.

Response:

Accepted.

[Figure]

**Supplement:**

[revised manuscript text omitted]

In the study, only three stations had another dominant land cover (site 1, 4 and 17). Especially for urban area that occupied much more part of the area, the sensible heat flux for non-vegetation type disaggregated from site 4 might be insufficient representative for the flux from sealed buildings and roads that are part of non-vegetation type. The divergence between modeled and measured flux may partly be attributed to this deficit.

Overall, the above results demonstrate that, compared with the area-averaged fluxes measured by LAS systems, the area-averaged fluxes that are aggregated from multiple EC flux measurements using the established flux aggregation method are reliable. 
[revised manuscript text omitted]

---

## Referee Report (RR1)

Area-averaged evapotranspiration over a heterogeneous land surface: Aggregation of multi-point EC flux measurements with high-resolution land-cover map and footprint analysis

Overall assessment:

The authors made substantial revisions to their manuscript. The revision is yet insufficient in several points. First, the present version failed to address the errors and their propagation, but simply stated that the errors in EC were corrected or minimized, done in published papers. It is unclear about the errors in the values aggregated from the area-averaged flux scheme they proposed. Second, the authors did not compare the scheme they proposed with those existing. It is hard to say the proposed scheme performed better than the others did. Third, the daily ET maps did not contribute to the central topics of the paper. Referee #3 also gave the similar comment.

Specific comments:

Line 8: avoid use of "etc." in research paper. I would not check grammatical problems. The manuscript requires a professional English editing.

Page 5 line 3: If you prefer to map daily ET over the whole study area. The daily values need validation. Otherwise, it is superficial.

Section 2.2: in the cases that errors in EC and LAS were clarified in published papers, please state it here in brief. Given the data is not error free, combined errors in aggregated flux should be addressed.

Page 25 line 10: the scheme relies on classification accuracy of land cover (inter-cover errors). It is also subject to seasonal change of any cover. Does it mean the regression coefficients are seasonally dependent?

Page 29 line 22 –page 30 line 19: Here compared area-averaged EC and LAS fluxes. The authors attributed the significant differences to residential areas. The reviewer agrees that residential areas are one of possible error sources, but this is not the whole story. As commented by referee#3, it is very necessary to clarify uncertainty in the EC and LAS measurements.

Section 4.4: Given that the large LAS-EC flux difference was not explained quantitatively, even for half an hour data, it is not rigorous to map the daily ET for the whole area.

---

## Author Response (AR3)

**A point-by-point response to the reviews**

First and foremost, we appreciate very much for the Editor and Referees' valuable and constructive comments on "Area-averaged evapotranspiration over a heterogeneous land surface: Aggregation of multi-point EC flux measurements with high-resolution land-cover map and footprint analysis" by F. Xu et al. According to Editor's comments, and those from

5   Referee #1 and Referee #4, we have carefully revised all sections of the paper (revisions and corrections are marked in red). Our detailed responses to their worthwhile comments and suggestions (Editor and referees' comments in italics) are as follows:

**Response to Editor's comments:**

**Main comments:**

10   *Three reviewers have provided some technical comments for your further consideration. I would like to ask you consider all of them and let me know how you considered them in your revision. In particular, I would like to suggest that you consider the following issues in the discussion.*

*1. The errors and their propagation in the area-averaged flux scheme.*

**Response:**

15   A detailed discussion on the errors induced from input data of present flux aggregation scheme, including the data of EC flux matrix, footprint model, land-cover map and LAS measurements, have been added in Sections 2.2 and 4.3. The standard errors of the EC dis-aggregated flux from least squares regression estimates were quantified and analyzed.

*2. Comparison of your proposed scheme to other existing in the literature.*

**Response:**

20   We have added comparisons between the area-weighted method and our proposed scheme with LAS measurements in Section 4.3. Fig. S1 shows some details of the results: in comparison with LAS measurements, the present scheme shows a rather less scatter than that of the area-weighted methods, where the 'simple area-averaged method' means that the sensible heat flux for each land cover class is simply the average of observed fluxes from the ECs in relevant land cover type; the 'area-averaged method' is that the sensible heat flux for each land cover class is the disaggregation of EC flux matrix by

25   present scheme.

[Figure]

**Figure S1:** Scatterplot of half-hourly area-averaged sensible heat fluxes estimated from flux aggregation method (H_ECagg) versus LAS measurements (H_LAS)

*3. The sensible heat flux retrieved from LAS is largely dependent on EC measurements. z0m, d, L, and Bowen ratio were estimated from EC, as is mentioned in the revised manuscript. A discussion should be given in how much this would influence the results of comparison between EC aggregation and LAS flux.*

**Response:**

A detailed discussion on this dependence has been added in Section 4.3. The flux uncertainties of LAS are influenced indeed by some parameters obtained from EC measurements, such as Obukhov length and Bowen ratio. Taking LAS 1 as a case (Fig. S2), when using a constant Bowen ratio in processing of the LAS data (as formerly did), the MAPE of the comparison between LAS and EC aggregation would be increased by about 20 %, and the value of RMSE would be more than 10 W m-2.

[Figure]

**Figure S2:** Comparison of area-averaged flux from EC flux matrix (H_ECagg) with that from LAS (H_LAS)

**Response to the comments from Referee #1:**

**Main comments:**

5  *1. The authors made substantial revisions to their manuscript. The revision is yet insufficient in several points. First, the present version failed to address the errors and their propagation, but simply stated that the errors in EC were corrected or minimized, done in published papers. It is unclear about the errors in the values aggregated from the area-averaged flux scheme they proposed.*

**Response:**

10  Thanks for your comments. As mentioned above, the systematic error of the turbulent flux from HiWATER flux matrix has been addressed in Section 2.2.1. A detailed discussion on the errors induced from input data of present flux aggregation scheme, including the data of EC flux matrix, footprint model, land-cover map and LAS measurements, has been added in Section 4.3. The standard errors of the EC dis-aggregated flux from least squares regression estimates were quantified and analyzed.

15  *2. Second, the authors did not compare the scheme they proposed with those existing. It is hard to say the proposed scheme performed better than the others did.*

**Response:**

As formerly stated, we have added comparisons between the area-weighted method and our proposed scheme with LAS

measurements in Section 4.3. Fig. S1 shows some details of the results: in comparison with LAS measurements, the present scheme shows a rather less scatter than that of the area-weighted methods.

*3. Third, the daily ET maps did not contribute to the central topics of the paper. Referee #3 also gave the similar comment.*

**Response:**

We have removed the daily ET maps that you and Referee #3 specified.

**Specific comments:**

*Line 8: avoid use of "etc." in research paper. I would not check grammatical problems. The manuscript requires a professional English editing.*

**Response:**

Accepted.

*Page 5 line 3: If you prefer to map daily ET over the whole study area. The daily values need validation. Otherwise, it is superficial.*

**Response:**

Thanks very much for your valuable comments. The daily ET maps in the manuscript have been dropped.

*Section 2.2: in the cases that errors in EC and LAS were clarified in published papers, please state it here in brief. Given the data is not error free. Combined errors in aggregated flux should be addressed.*

**Response:**

Accepted.

The systematic error of the turbulent flux from HiWATER flux matrix was 8 %, which was indirectly obtained through the surface energy balance closure (EBR). Details have been stated in Section 2.2.1. The uncertainties of H and LE over maize fields in the kernel experimental area were approximately 18 % and 16 %, respectively (Wang et al., 2015). Besides, the errors in aggregated flux have been addressed in Section 4.3.

*Page 25 line 10: the scheme relies on classification accuracy of land cover (inter-cover errors). It is also subject to seasonal change of any cover. Does it mean the regression coefficients are seasonally dependent?*

**Response:**

Yes. Two clear days we selected for analysis are typical, due to the weather, surface status and extended observations. The main aim is to establish a flux aggregation scheme and check its reliability. In the future work, we will continue to explore the relationship between the regression coefficients and seasonal change of any cover. After that, we are going to obtain long-term disaggregated ET for the validation of remote sensing ET products. And this work will be extended to the water balance study of the whole Heihe River basin.

*Page 29 line 22- page 30 line 19: Here compared area-averaged EC and LAS fluxes. The authors attributed the significant differences to residential areas. The reviewer agrees that residential areas are one of possible error sources, but this is not the whole story. As commented by referee #3, it is very necessary to clarify uncertainty in the EC and LAS measurements.*

**Response:**

Accepted. Some statements about the effect of uncertainty in EC and LAS measurements on the comparison of EC aggregated area-averaged fluxes and LAS measured fluxes have been added into Section 4.3.

*Section 4.4: Given that the large LAS-EC flux difference was not explained quantitatively, even for half an hour data, it is not rigorous to map the daily ET for the whole area.*

**Response:**

Thanks very much for your valuable comments. Accepted.

**Response to the comments from Referee #4:**

**Main comments:**

*Please reduce the accuracy of the data, e.g. 20.61 Wm-2 to 21 Wm-2 or 19.31 % to 19 %. The accuracy of the presented data should be in agreement with the accuracy of the fluxes etc. that was discussed in the paper.*

**Response:**

Accepted.

**A marked-up manuscript version**

[revised manuscript text omitted]